# Fine-grained Analysis of In-context Linear Estimation:
# Data, Architecture, and Beyond

**Yingcong Li** [1]   **Ankit Singh Rawat** [2]   **Samet Oymak** [1]

## Abstract

Recent research has shown that Transformers with linear attention are capable of in-context learning (ICL) by implementing a linear estimator through gradient descent steps. However, the existing results on the optimization landscape apply under stylized settings where task and feature vectors are assumed to be IID and the attention weights are fully parameterized. In this work, we develop a stronger characterization of the optimization and generalization landscape of ICL through contributions on architectures, low-rank parameterization, and correlated designs: (1) We study the landscape of 1-layer linear attention and 1-layer H3, a state-space model. Under a suitable correlated design assumption, we prove that both implement 1-step preconditioned gradient descent. We show that thanks to its native convolution filters, H3 also has the advantage of implementing sample weighting and outperforming linear attention in suitable settings. (2) By studying correlated designs, we provide new risk bounds for retrieval augmented generation (RAG) and task-feature alignment which reveal how ICL sample complexity benefits from distributional alignment. (3) We derive the optimal risk for low-rank parameterized attention weights in terms of covariance spectrum. Through this, we also shed light on how LoRA can adapt to a new distribution by capturing the shift between task covariances. Experimental results corroborate our theoretical findings. Overall, this work explores the optimization and risk landscape of ICL in practically meaningful settings and contributes to a more thorough understanding of its mechanics.

[1]University of Michigan [2]Google Research NYC. Correspondence to: Samet Oymak <oymak@umich.edu>.

*Proceedings of the 1st Workshop on In-Context Learning at the 41st International Conference on Machine Learning*, Vienna, Austria. 2024. Copyright 2024 by the author(s).

## 1. Introduction

Modern language models exhibit remarkable ability to learn novel tasks or solve complex problems from the demonstrations provided within their context window (Brown et al., 2020; GeminiTeam et al., 2023; OpenAI, 2023; Touvron et al., 2023). Such *in-context learning* (ICL) offers a novel and effective alternative to traditional fine-tuning techniques. It enables successful prediction across a wide range of tasks simply through a forward pass, eliminating the need for task-specific model weight updates. Since its introduction, ICL capability has become an important feature of LLM with its applications spanning retrieval-augmented generation (Lewis et al., 2020), and reasoning via advanced prompting techniques, such as chain-of-thought (Wei et al., 2022). While ICL already exhibits considerable benefits with a small number of demonstrations, i.e., few-shot data, there is a growing interest in extending its benefits to the many-shot settings, potentially realizing even more pronounced benefits (Agarwal et al., 2024).

ICL ability also presents an important research avenue to develop stronger theoretical and mechanistic understanding of large language models. To this aim, there has been significant recent interest in demystifying ICL through the lens of function approximation (Liu et al., 2023a), Bayesian inference (Müller et al., 2021; Xie et al., 2022; Han et al., 2023), and learning and optimization theory (Ahn et al., 2023; Mahankali et al., 2024; Zhang et al., 2024; Duraisamy, 2024). The latter is concerned with understanding the optimization landscape of ICL, which is also crucial for understanding the generalization properties of the model. A notable result in this direction is the observation that linear attention models (Schlag et al., 2021; Von Oswald et al., 2023; Ahn et al., 2023) implement *preconditioned gradient descent* (PGD) during ICL (Ahn et al., 2023; Mahdavi et al., 2024). While this line of works provide a fresh perspective to ICL, the existing studies do not address many questions arising from real-life applications nor provide guiding principles for various ICL setups motivated by practical considerations.

To this aim, we revisit the theoretical exploration of ICL with linear data model where we feed an in-context prompt containing $n$ examples $(\boldsymbol{x}_i, y_i = \boldsymbol{x}_i^\top \boldsymbol{\beta} + \xi_i)_{i=1}^n \subset \mathbb{R}^d \times \mathbb{R}$ and a test instance or query $\boldsymbol{x}_{n+1} \in \mathbb{R}^d$ to the model, where $d$ is

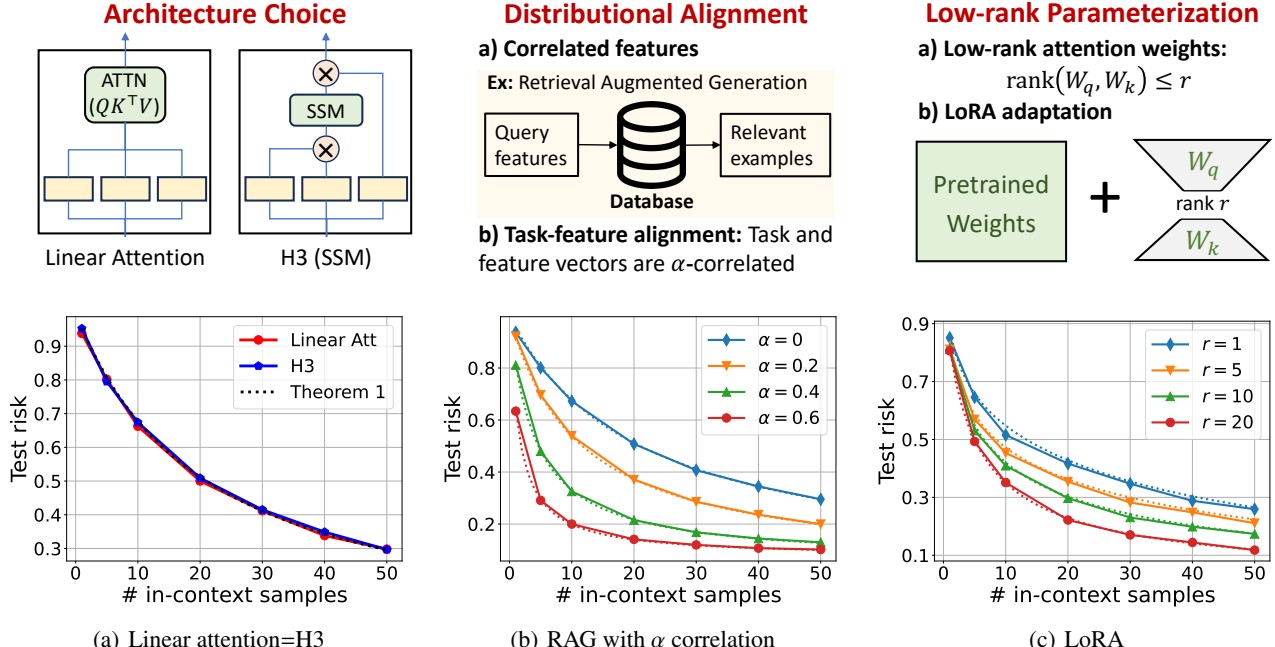

(a) Linear attention=H3  (b) RAG with $\alpha$ correlation  (c) LoRA

Figure 1: We investigate the optimization landscape of in-context learning from the lens of architecture choice, the role of distributional alignment, and low-rank parameterization. The empirical performance (solid curves) are aligned with our theoretical results (dotted curves) from Section 3. More experimental details and discussion are deferred to Section 4.

the feature dimension, $\boldsymbol{\beta} \in \mathbb{R}^d$ is the task vector, and $(\xi_i)_{i=1}^n$ denote the noise in individual labels. Given the in-context prompt, the model is tasked to predict $\hat{y}_{n+1}$ – an estimate for $y_{n+1} = \boldsymbol{x}_{n+1}^\top \boldsymbol{\beta} + \xi_{n+1}$. We aim to provide answers to the following questions by exploring the loss landscape of ICL:

(Q1) Is the ability to implement gradient-based ICL unique to (linear) attention? Can alternative sequence models implement richer algorithms beyond PGD?

(Q2) In language modeling, ICL often works well with few-shot samples whereas standard linear estimation typically requires $O(d)$ samples. How to reconcile this discrepancy between classical learning and ICL?

(Q3) To our knowledge, existing works assume linear-attention is fully parameterized, i.e., key and query projections $\boldsymbol{W}_k, \boldsymbol{W}_q \in \mathbb{R}^{d \times d}$. What happens when they are low-rank? What happens when there is distribution shift between training and test in-context prompts and we use LoRA (Hu et al., 2022) for adaptation?

In this work, we conduct a careful investigation of these questions. Specifically, we focus on ICL with 1-layer models and make the following contributions:

(A1) We jointly investigate the landscape of linear attention and H3 (Fu et al., 2023), a widely popular state-space model (SSM). We prove that under correlated design, both models implement 1-step PGD (c.f. Proposition 2.3) and the alignments in Fig. 1(a) verify that

where the dotted curve represents the theoretical result derived from Theorem 3.1. Our analysis reveals that the gating mechanism in H3 imitates attention. We also empirically show that H3 has the advantage of implementing sample-weighting which allows it to outperform linear attention in temporally-heterogeneous problem settings in Section 4 and Figure 4.

(A2) Proposition 2.3 allows for task and features to be correlated to each other as long as odd moments are zero. Through this, we can assess the impact of distributional alignment on the sample complexity of ICL. Specifically, we characterize the performance of *Retrieval Augmented Generation* (RAG) (c.f. Theorem 3.3 and Fig. 1(b)) and *Task-Feature Alignment* (c.f. Theorem 3.4), where the in-context examples are $\alpha$-correlated with either the query or the task vector. For both settings, we prove that alignment amplifies the *effective sample size* of ICL by a factor of $\alpha^2 d + 1$, highlighting that aligned data are crucial for the success of ICL in few-shot settings.

(A3) We show that, under low-rank parameterization, optimal attention-weights still implements PGD according to the truncated eigenspectrum of the fused task-feature covariance (see Section 3.2). We similarly derive risk upper bounds for LoRA adaptation (c.f. Eq. (14) and Fig. 1(c)), and show that, these bounds accurately predict the empirical performance.

## 2. Problem Setup and Preliminaries

We begin with a short note on notation. Let bold lower-case and uppercase letters (e.g., $x$ and $X$) represent vectors and matrices, respectively. The symbol $\odot$ is defined as the element-wise (Hadamard) product, and $*$ denotes the convolution operator. $\mathbf{1}_d$ and $\mathbf{0}_d$ denote the $d$-dimensional all-ones and all-zeros vectors, respectively; and $I_d$ denotes the identity matrix of dimension $d \times d$. Additionally, let $\mathtt{tr}(W)$ denote the trace of the square matrix $W$.

As mentioned earlier, we study the optimization landscapes of 1-layer linear attention (Katharopoulos et al., 2020; Schlag et al., 2021) and H3 (Fu et al., 2023) models when training with prompts containing in-context data following a linear model. We construct the input in-context prompt similar to Ahn et al. (2023); Mahankali et al. (2024); Zhang et al. (2024) as follows.

**Linear data distribution.** Let $(x, y) \in \mathbb{R}^d \times \mathbb{R}$ be a (feature, label) pair generated by a $d$-dimensional linear model parameterized by $\beta \in \mathbb{R}^d$, i.e., $y = x^\top \beta + \xi$, where $x$ and $\beta$ are feature and task vectors, and $\xi$ is the label noise. Given demonstrations $(x_i, y_i)_{i=1}^{n+1}$ sampled from a single $\beta$, define the input in-context prompt

$$Z = [z_1 \ \ldots \ z_n \ z_{n+1}]^\top = \begin{bmatrix} x_1 & \ldots & x_n & x_{n+1} \\ y_1 & \ldots & y_n & 0 \end{bmatrix}^\top. \quad (1)$$

Here, we set $z_i = \begin{bmatrix} x_i \\ y_i \end{bmatrix}$ for $i \leq n$ and the last/query token $z_{n+1} = \begin{bmatrix} x_{n+1} \\ 0 \end{bmatrix}$. Then, given $Z$, the goal of the model is to predict the correct label $y_{n+1}$ corresponding to $x_{n+1}$. For cleaner notation, when it is clear from context, we drop the subscript $n + 1$ and set $x = x_{n+1}$, $z = z_{n+1}$. Different from the previous work (Ahn et al., 2023; Mahankali et al., 2024; Zhang et al., 2024; Mahdavi et al., 2024) where $(x_i)_{i=1}^{n+1}$ and $\beta$ are assumed to be independent, our analysis focuses on a more general linear setting that captures the dependency between $(x_i)_{i=1}^{n+1}$ and $\beta$.

**Model architectures.** To start with, we first review the architectures of both Transformer and state-space model (SSM). Similar to the previous work (Von Oswald et al., 2023; Ahn et al., 2023; Mahankali et al., 2024; Zhang et al., 2024) and to simplify the model structure, we focus on single-layer models and omit the nonlinearity, e.g., softmax operation and MLP activation, from the Transformer. Given the input prompt $Z \in \mathbb{R}^{(n+1)\times(d+1)}$ in (1), which can be treated as a sequence of $(d + 1)$-dimensional tokens, the single-layer linear attention $\mathtt{ATT}$ and H3-like single-layer SSM $\mathtt{SSM}$ are denoted by

$$\mathtt{ATT}(Z) = (ZW_q W_k^\top Z^\top) Z W_v \quad (2a)$$

$$\mathtt{SSM}(Z) = \big((ZW_q) \odot ((ZW_k \odot ZW_v) * f)\big) \quad (2b)$$

where $W_k$, $W_q$, $W_v \in \mathbb{R}^{(d+1)\times(d+1)}$ denote the key, query and value weight matrices, respectively. In (2b), the parameter $f \in \mathbb{R}^{n+1}$ is a 1-D convolutional filter that mixes tokens. The Hadamard product $\odot$ is the gating mechanism (Dauphin et al., 2017) between key and query channels, which is crucial for attention-like feature creation. Thus, (2b) is more generally a gated-convolution layer. For $f$ only, we use indexing $f = [f_0 \ \ldots \ f_n]^\top \in \mathbb{R}^{n+1}$ and given any vector $a$, denote convolution output $(a * f)_i = \sum_{j=1}^i f_{i-j} a_j$. Note that our notation slightly differs from the original H3 model (Fu et al., 2023) in two ways:

1. SSMs provide efficient parameterization of $f$ which would otherwise grow with sequence length. In essence, H3 utilizes a linear state-space model $s_i = As_{i-1} + Bu_i$ and $y_i = Cs_i$ with parameters ($A \in \mathbb{R}^{d\times d}, B \in \mathbb{R}^{d\times 1}, C \in \mathbb{R}^{1\times d}$) from which the filter $f$ is obtained via the impulse response $f_i = CA^iB$ for $i \geq 0$. Here $d$ is the state dimension and, in practice, $A$ is chosen to be diagonal. Observe that, setting $d = 1$ and $A = \rho, C = B = 1$, SSM reduces to the exponential smoothing $f_i = \rho^i$ for $i \geq 0$. Thus, H3 also captures the all-ones filter as a special instance. As we show in Proposition 2.3, this simple filter is optimal under independent data model and exactly imitates linear attention. Note that, utilizing a filter $f$ as in (2b) is strictly more expressive than the SSM as it captures all possible impulse responses.

2. H3 also applies a shift SSM to the key embeddings to enable the retrieval of the local context around associative recall hits. We opted not to incorporate this shift operator in our model. This is because unless the features of the neighboring tokens are correlated (which is not the case for the typical independent data model), the entry-wise products between values and shifted keys will have zero mean and be redundant for the final prediction.

We note that we conduct all empirical evaluations with the original H3 model, which displays exact agreement with our theory formalized for (6b), further validating our modeling choice.

### 2.1. In-context Linear Estimation

We will next study the algorithms that can be implemented by the single-layer attention and state-space models. Through this, we will show that training $\mathtt{ATT}$ and $\mathtt{SSM}$ with linear ICL data is equivalent to the prediction obtained from one step of optimally-*preconditioned gradient descent* (PGD) and *sample-weighted preconditioned gradient descent* (WPGD), respectively. We will further show that under mild assumption, the optimal sample weighting for

SSM (e.g., $f$) is an all-ones vector and therefore, establishing the equivalence among PGD, ATT, and SSM.

**Background: 1-step gradient descent.** Consider minimizing squared loss and solving linear regression using one step of PGD and WPGD. Given $n$ samples $(x_i, y_i)_{i=1}^n$, define

$$X = [x_1 \; \cdots \; x_n]^\top \in \mathbb{R}^{n \times d} \quad \text{and} \quad y = [y_1 \; \cdots \; y_n]^\top \in \mathbb{R}^n.$$

Starting from $\beta_0 = \mathbf{0}_d$ and letting $\eta = 1/2$ be the step size, 1-step GD preconditioned with weights $W$ returns prediction

$$\hat{y} = x^\top W X^\top y := g_{\text{PGD}}(Z), \tag{3}$$

and a single-step *sample-weighted* GD given weights $\omega \in \mathbb{R}^n$ and $W \in \mathbb{R}^{d \times d}$ returns prediction

$$\hat{y} = x^\top W X^\top (\omega \odot y) := g_{\text{WPGD}}(Z), \tag{4}$$

where $Z$ is defined in (1) consisting of $X, y$ and $x$. Our goal is to find the optimal $W$, as well as $\omega$ in (4) that minimize the population risks defined as follows.

$$\min_W \mathcal{L}_{\text{PGD}}(\mathcal{W}) \text{ where } \mathcal{L}_{\text{PGD}}(\mathcal{W}) = \mathbb{E}\left[(y - g_{\text{PGD}}(Z))^2\right], \tag{5a}$$

$$\min_{W,\omega} \mathcal{L}_{\text{WPGD}}(\mathcal{W}) \text{ where } \mathcal{L}_{\text{WPGD}}(\mathcal{W}) = \mathbb{E}\left[(y - g_{\text{WPGD}}(Z))^2\right]. \tag{5b}$$

Here, the expectation is over the randomness in $(x_i, \xi_i)_{i=1}^{n+1}$ and $\beta$, and we use $\mathcal{W}$ to represent the set of corresponding trainable parameters. The search spaces for $\omega$ and $W$ are $\mathbb{R}^n$ and $\mathbb{R}^{d \times d}$, respectively.

As per (2), given input prompt $Z \in \mathbb{R}^{(n+1) \times (d+1)}$, either of the underlying models outputs a $(n+1)$-length sequence. Note that the label for the query $x = x_{n+1}$ is excluded from the prompt $Z$. Similar to Ahn et al. (2023); Mahankali et al. (2024), we consider a training objective with a causal mask to ensure inputs cannot attend to their own labels and training can be parallelized. Let $Z_0 = [z_1 \; \ldots \; z_n \; 0]^\top$ be the features post-causal masking at time/index $n+1$. Given weights $W_k, W_q, W_v$ and the filter $f$ for SSM, predictions at the query token $z = \begin{bmatrix} x \\ 0 \end{bmatrix}$ take the following forms following sequence-to-sequence mappings in (2):

$$g_{\text{ATT}}(Z) = (z^\top W_q W_k^\top Z_0^\top) Z_0 W_v v,$$

$$g_{\text{SSM}}(Z) = \left((z^\top W_q)^\top \odot ((Z_0 W_k \odot Z_0 W_v) * f)_{n+1}\right) v,$$

where $v \in \mathbb{R}^{d+1}$ is the linear prediction head and $((Z_0 W_k \odot Z_0 W_v) * f)_{n+1}$ returns the last row of the convolution output. Note that SSM can implement the mask by setting $f_0 = 0$. Now consider the meta learning setting and select loss function to be the squared loss, same as in (5). Thus, the objectives for both models take the following forms.

$$\min_{W_k, W_q, W_v, v} \mathcal{L}_{\text{ATT}}(\mathcal{W}) \text{ where } \mathcal{L}_{\text{ATT}}(\mathcal{W}) = \mathbb{E}\left[(y - g_{\text{ATT}}(Z))^2\right], \tag{6a}$$

$$\min_{W_k, W_q, W_v, v, f} \mathcal{L}_{\text{SSM}}(\mathcal{W}) \text{ where } \mathcal{L}_{\text{SSM}}(\mathcal{W}) = \mathbb{E}\left[(y - g_{\text{SSM}}(Z))^2\right]. \tag{6b}$$

Here, similarly, the expectation subsumes the randomness of $(x_i, \xi_i)_{i=1}^{n+1}$ and $\beta$ and $\mathcal{W}$ represents the set of trainable parameters. The search space for matrices $W_k, W_q, W_v$ is $\mathbb{R}^{(d+1) \times (d+1)}$, for head $v$ is $\mathbb{R}^{d+1}$, and for $f$ is $\mathbb{R}^{n+1}$.

Note that for all the optimization methods (c.f. (5), (6)), to simplify the analysis, we train the models without capturing additional bias terms. Therefore, in the following, we introduce the centralized data assumptions such that the models are trained to make unbiased predictions.

To begin with, a cross moment of random variables is defined as the expectation of a monomial of these variables, with the order of the cross moment being the same as order of the monomial. For example, $\mathbb{E}[x^\top W \beta]$ is a sum of cross-moments of order 2. Then, it motivates the following data assumptions.

**Assumption 2.1.** All cross moments of the entries of $(x_i)_{i=1}^{n+1}$ and $\beta$ with odd orders are zero.

**Assumption 2.2.** $(\xi_i)_{i=1}^{n+1}$ are independent of $(x_i)_{i=1}^{n+1}$ and $\beta$, and their cross moments with odd orders are zero.

Note that compared to (Ahn et al., 2023; Mahankali et al., 2024; Zhang et al., 2024), Assumption 2.1 is more general which also subsumes the dependent distribution settings. In this work, we consider the following three linear models (omitting noise) satisfying Assumption 2.1. Let $\Sigma_\beta, \Sigma_x \in \mathbb{R}^{d \times d}$ represent the task and feature covariance matrices for independent data, and let $0 \le \alpha \le 1$ be the correlation level when considering data dependency. More specific discussions are deferred to Section 3.

- Independent task and data:

$$\beta \sim \mathcal{N}(0, \Sigma_\beta), \; x_i \sim \mathcal{N}(0, \Sigma_x), \; i \le n + 1.$$

- Retrieval augmented generation:

$$\beta, x \sim \mathcal{N}(0, I_d), \; x_i \,|\, x \sim \mathcal{N}(\alpha x, (1 - \alpha^2) I_d), \; i \le n.$$

- Task-feature alignment:

$$\beta \sim \mathcal{N}(0, I_d), \; x_i \,|\, \beta \sim \mathcal{N}(\alpha \beta, I_d), \; i \le n + 1.$$

Next, we introduce the following result which establishes the equivalence among optimizing linear attention (c.f. (6a)), H3 (c.f. (6b)), and gradient descent (c.f. (5)).

**Proposition 2.3.** *Suppose Assumptions 2.1 and 2.2 hold. Consider the objectives as defined in* (5) *and* (6), *and let* $\mathcal{L}_{\text{PGD}}^\star, \mathcal{L}_{\text{WPGD}}^\star, \mathcal{L}_{\text{ATT}}^\star$, *and* $\mathcal{L}_{\text{SSM}}^\star$ *be their optimal risks, respectively. Then,*

$$\mathcal{L}_{\text{PGD}}^\star = \mathcal{L}_{\text{ATT}}^\star \quad and \quad \mathcal{L}_{\text{WPGD}}^\star = \mathcal{L}_{\text{SSM}}^\star.$$

*Additionally, if the examples* $(x_i, y_i)_{i=1}^n$ *follow the same distribution and are conditionally independent given* $x, \beta$, *then SSM/H3 can achieve the optimal loss using the all-ones filter and* $\mathcal{L}_{\text{PGD}}^\star = \mathcal{L}_{\text{SSM}}^\star$.

We defer the proof to Appendix B.1. Proposition 2.3 establishes that analyzing the optimization landscape of ICL for both single-layer linear attention and the H3 model can be effectively reduced to examining the behavior of a one-step PGD algorithm. Notably, under the independent, RAG and task-feature alignment data settings discussed above, examples $(x_i, y_i)_{i=1}^n$ are independently sampled given $x$ and $\beta$, and we therefore conclude that $\mathcal{L}_{\text{PGD}}^\star = \mathcal{L}_{\text{ATT}}^\star = \mathcal{L}_{\text{SSM}}^\star$. Leveraging this result, the subsequent section of the paper concentrate on addressing (5a), taking into account various linear data distributions.

While Proposition 2.3 demonstrates the equivalence of optimal losses, we also study the uniqueness and equivalence of optimal prediction functions. To this end, we analyze the strong convexity of $\mathcal{L}_{\text{PGD}}(\mathcal{W})$.

**Lemma 2.4.** *Suppose Assumption 2.2 holds and let $\xi = [\xi_1 \, \xi_2 \, \cdots \, \xi_n]^\top$. Then the loss $\mathcal{L}_{\text{PGD}}(\mathcal{W})$ in (5a) is strongly-convex iff. $\mathbb{E}[(x^\top W X^\top X \beta)^2] + \mathbb{E}[(x^\top W X^\top \xi)^2]$ is strongly-convex. Additionally, let $g_{\text{PGD}}^\star$, $g_{\text{ATT}}^\star$ be the optimal prediction functions of (5a), (6a). Then under the conditions of Assumptions 2.1 and 2.2, and the strong convexity, $g_{\text{PGD}}^\star = g_{\text{ATT}}^\star$.*

**Lemma 2.5.** *Suppose that the label noise $(\xi_i)_{i=1}^n$ are i.i.d., zero-mean, variance $\sigma^2$ and independent of everything else, and that there is a decomposition $x = x_1 + x_2$, $X = X_1 + X_2$, and $\beta = \beta_1 + \beta_2$ such that either of the following holds*

- *$\sigma > 0$, and $(x_1, X_1)$ have full rank covariance and are independent of each other and $(x_2, X_2)$.*

- *$(x_1, \beta_1, X_1)$ have full rank covariance and are independent of each other and $(x_2, \beta_2, X_2)$.*

*Then, the loss $\mathcal{L}_{\text{PGD}}(\mathcal{W})$ in (5a) is strongly-convex.*

As mentioned above, we study three specific linear models: with general independent, RAG-related, and task-feature alignment data. Note that for all the three cases, according to Proposition 2.3, we have $\mathcal{L}_{\text{PGD}}^\star = \mathcal{L}_{\text{ATT}}^\star = \mathcal{L}_{\text{SSM}}^\star$. Additionally, the second claim in Lemma 2.5 holds, and $\mathcal{L}_{\text{PGD}}(\mathcal{W})$ is strongly convex. Therefore, following Lemma 2.4, we have $g_{\text{PGD}}^\star = g_{\text{ATT}}^\star$. Thanks to the equivalence among PGD, ATT, and SSM, in the next section, we focus on the solution of objective (5a) under different scenarios, which will reflect the optimization landscapes of ATT and SSM models.

## 3. Main Results

In light of Proposition 2.3, optimizing a single layer linear-attention or H3 model is equivalent to solving the objective (5a). Therefore, in this section, we examine the properties of the one-step PGD in (5a). To this end, we consider multiple problem settings, including distinct data distributions and low-rank training.

### 3.1. Analysis of Linear Data Models

We first consider the standard independent data setting. We will then examine correlated designs.

**Independent data model.** Let $\Sigma_x$ and $\Sigma_\beta$ be the covariance matrices of the input feature and task vectors, respectively, and $\sigma \geq 0$ be the noise level. We assume

$$\beta \sim \mathcal{N}(0, \Sigma_\beta), \; x_i \sim \mathcal{N}(0, \Sigma_x), \; \xi_i \sim \mathcal{N}(0, \sigma^2), \; i \leq n + 1 \quad (7)$$

and the label is obtained via $y_i = x_i^\top \beta + \xi_i$. Our following result characterizes the optimal solution of (5a). Note that the data generated from (7) satisfies the conditions in Proposition 2.3. Therefore, the same results can be applied to both linear-attention and H3 models.

**Theorem 3.1.** *Consider independent linear data in (7), and suppose the covariance matrices $\Sigma_x, \Sigma_\beta$ are full rank. Recap the objective from (5a) and let $W_\star := \arg\min_W \mathcal{L}_{\text{PGD}}(W)$, and $\mathcal{L}_\star = \mathcal{L}_{\text{PGD}}(W_\star)$. Additionally, let $\Sigma = \Sigma_x^{1/2} \Sigma_\beta \Sigma_x^{1/2}$ and $M = \text{tr}(\Sigma) + \sigma^2$. Then $W_\star$ and $\mathcal{L}_\star$ satisfy*

$$W_\star = \Sigma_x^{-1/2} \bar{W}_\star \Sigma_x^{-1/2} \quad and \quad \mathcal{L}_\star = M - n\text{tr}(\Sigma \bar{W}_\star), \quad (8)$$

*where we define $\bar{W}_\star = \left((n+1)I_d + M\Sigma^{-1}\right)^{-1}$.*

**Corollary 3.2.** *Consider noiseless i.i.d. linear data where $\Sigma_x = \Sigma_\beta = I_d$ and $\sigma = 0$. Then, the objective in (5a) returns*

$$W_\star = \frac{1}{n+d+1} I_d \quad and \quad \mathcal{L}_\star = d - \frac{nd}{n+d+1}.$$

See Appendix C.2 for proofs. Note that Theorem 3.1 is consistent with prior work (Ahn et al., 2023, Theorem 1) when specialized to isotropic task covariance, i.e., $\Sigma_\beta = I_d$. However, their result is limited as the features and task are assumed to be independent. This prompts us to ask: *What is the optimization landscape with correlated in-context samples?* Toward this, we consider the following RAG-inspired and task-feature alignment models, where Assumptions 2.1 and 2.2 continue to hold and Proposition 2.3 applies.

**Retrieval augmented generation.** To provide a statistical model of the practical RAG approaches, given the query vector $x_{n+1} = x$, we propose to draw ICL demonstrations that are similar to $x$ with the same shared task vector $\beta$. Modeling feature similarity through the cosine angle, RAG should sample the ICL examples $x_i, i \leq n$, from the original feature distribution conditioned on the event $\cos(x_i, x) \geq \alpha$ where $\alpha$ is the similarity threshold. As an approximate proxy, under the Gaussian distribution model, we assume that $\beta \sim \mathcal{N}(0, I_d)$, $x \sim \mathcal{N}(0, I_d)$ and that RAG samples $\alpha$-correlated demonstrations $(x_i, y_i)_{i=1}^n$ as follows:

$$x_i \mid x \sim \mathcal{N}(\alpha x, (1 - \alpha^2) I_d), \; \xi_i \sim \mathcal{N}(0, \sigma^2), \; 1 \leq i \leq n \quad (9)$$

and $y_i = x_i^\top \beta + \xi_i$. Note that the above normalization ensures that the marginal feature distribution remains $\mathcal{N}(0, I_d)$.

The full analysis of RAG is provides in Appendix C.3. Specifically, when we carry out the analysis by assuming $\alpha = O\left(1/\sqrt{d}\right)$ and $d/n = O(1)$ where $O(\cdot)$ denotes proportionality, our derivation leads to the following result:

**Theorem 3.3.** *Consider linear model in* (9). *Recap the objective from* (5a) *and let* $W_\star := \arg\min_W \mathcal{L}_{PGD}(W)$, *and* $\mathcal{L}_\star = \mathcal{L}_{PGD}(W_\star)$. *Additionally, let* $\kappa = \alpha^2 d + 1$ *and suppose* $\alpha = O\left(1/\sqrt{d}\right)$, $d/n = O(1)$ *and* $d$ *is sufficiently large. Then* $W_\star$ *and* $\mathcal{L}_\star$ *have approximate forms*

$$W_\star \approx \frac{1}{\kappa n + d + \sigma^2} I_d \ \text{ and } \ \mathcal{L}_\star \approx d + \sigma^2 - \frac{\kappa n d}{\kappa n + d + \sigma^2}. \quad (10)$$

Here, (10) is reminiscent of Corollary 3.2 and has a surprisingly clean message. Observe that, $\alpha^2 d + 1$ is the dominant multiplier ahead of $n$ in both equations. Thus, we deduce that, RAG model follows the same error bound as the independent data model, however, its sample size is amplified by a factor of $\alpha^2 d + 1$. $\alpha = 0$ reduces to the result of Corollary 3.2 whereas we need to set $\alpha = O\left(1/\sqrt{d}\right)$ for constant amplification. When $\alpha = 1$, RAG achieves the approximate risk $\mathcal{L}_\star \approx 2 + \sigma^2$, where the constant bias is due to the higher order moments (e.g., the 4'th and 6'th moments) of the standard Gaussian distribution. As $d$ increases, the normalized loss $\mathcal{L}_\star/d \to 0$. The full analysis of its optimal solution $W_\star$ and loss $\mathcal{L}_\star$ are deferred Theorem C.1 in Appendix C.3.

**Task-feature alignment.** We also consider another dependent data setting where task and feature vectors are assumed to be correlated. This dataset model has the following motivation: In general, an LLM can generate any token within the vocabulary. However, once we specify the task (e.g. domain of the prompt), the LLM output becomes more deterministic and there are much fewer token candidates. For instance, if the task is "Country", "France" is a viable output compared to "Helium" and vice versa when the task is "Chemistry". Formally speaking, this can be formalized as the input $x$ having a diverse distribution whereas it becomes more predictable conditioned on $\beta$. Therefore, it can be captured through a linear model by making the conditional covariance of $x \mid \beta$ to be approximately low-rank. This formalism can be viewed as a *spectral alignment* between input and task, which is also well-established in deep learning both empirically and theoretically (Li et al., 2020; Arora et al., 2019; Canatar et al., 2021; Cao et al., 2019). Here, we consider such a setting where the shared task vector is sampled as standard Gaussian distribution $\beta \sim \mathcal{N}(0, I_d)$ and letting $\kappa = \alpha^2 d + 1$, we sample the $\alpha$-correlated ICL demonstrations $(x_i, y_i)_{i=1}^{n+1}$ as follows:

$$x_i \mid \beta \sim \mathcal{N}(\alpha\beta, I_d), \ \xi_i \sim \mathcal{N}(0, \sigma^2), \ 1 \le i \le n+1. \quad (11)$$

and $y_i = \kappa^{-1/2} x_i^\top \beta + \xi_i$. Here, $\kappa^{-1/2}$ is a normalization factor to ensure that label variance remains invariant to $\alpha$. To keep the exposition cleaner, we defer the full analysis of

its optimal solution $W_\star$ and loss $\mathcal{L}_\star$ to Theorem C.2 in Appendix C.4. Similar to the RAG setting, by assuming $\alpha = O\left(1/\sqrt{d}\right)$ and $d/n = O(1)$, we obtain the following results for the optimal parameter and risk.

**Theorem 3.4.** *Consider linear model in* (11). *Recap the objective from* (5a) *and let* $W_\star := \arg\min_W \mathcal{L}_{PGD}(W)$, *and* $\mathcal{L}_\star = \mathcal{L}_{PGD}(W_\star)$. *Additionally, given* $\kappa = \alpha^2 d + 1$ *and suppose* $\alpha = O\left(1/\sqrt{d}\right)$, $d/n = O(1)$ *and* $d$ *is sufficiently large. Then* $W_\star$ *and* $\mathcal{L}_\star$ *have approximate forms*

$$W_\star \approx \frac{1}{\kappa n + (d + \sigma^2)/\kappa} I_d \ \text{and} \ \mathcal{L}_\star \approx d + \sigma^2 - \frac{\kappa n d}{\kappa n + (d + \sigma^2)/\kappa}. \quad (12)$$

Similar to (10), (12) contains $\kappa = \alpha^2 + 1$ multiplier ahead of $n$, which reduces the in-context sample complexity and setting $\alpha = 0$ reduces to the results of Corollary 3.2.

### 3.2. Low-rank Parameterization and LoRA

In this section, we investigate training low-rank models, which assume $W_k, W_q \in \mathbb{R}^{(d+1)\times r}$ where $r$ is the rank restriction. Equivalently, we consider objective (5a) under condition $\text{rank}(W) = r$.

**Lemma 3.5.** *Consider independent linear data in* (7). *Recap the objective from* (5a) *and enforce* $\text{rank}(W) \le r$ *and* $W^\top = W$. *Let* $\Sigma = \Sigma_x^{1/2} \Sigma_\beta \Sigma_x^{1/2}$ *and* $M = \text{tr}(\Sigma) + \sigma^2$. *Let* $\lambda_i$ *be the* $i$'th *largest eigenvalue of* $\Sigma$, *we have that*

$$\min_{\text{rank}(W)\le r, W=W^\top} \mathcal{L}(W) = M - \sum_{i=1}^{r} \frac{n\lambda_i^2}{(n+1)\lambda_i + M}. \quad (13)$$

Note that $\text{tr}(\Sigma) = \sum_{i=1}^{d} \lambda_i$. Removing the rank constraint and considering noiseless data setting, this reduces to the following optimal risk $\mathcal{L}_\star = \sum_{i=1}^{d} \frac{\lambda_i + M}{n+1+M/\lambda_i}$. See Appendix D.1 for more details.

**Impact of LoRA:** Based on the above lemma, we consider the impact of LoRA for adapting the pretrained model to a new task distribution under jointly-diagonalizable old and new eigenvalues of $\Sigma$, $\Sigma^{new}$, $(\lambda_i)_{i=1}^{d}$, $(\lambda_i^{new})_{i=1}^{d}$. Consider adapting LoRA matrix to the combined key and value weights in attention, which reflects minimizing the population loss $\tilde{\mathcal{L}}(W_{lora}) := \mathcal{L}(W + W_{lora})$ in (5a) with fixed $W$. Suppose $\text{tr}(\Sigma) = \text{tr}(\Sigma^{new}) = M$, $\sigma = 0$ and $W$ is jointly diagonalizable with $\Sigma$, $\Sigma^{new}$, then LoRA's risk is upper-bounded by

$$\min_{\text{rank}(W_{lora})\le r} \tilde{\mathcal{L}}(W_{lora}) \le$$
$$\min_{|\mathcal{I}|\le r, \mathcal{I}\subset[d]} \left( \sum_{i\notin\mathcal{I}} \frac{\lambda_i + M}{n+1+M/\lambda_i} + \sum_{i\in\mathcal{I}} \frac{\lambda_i^{new} + M}{n+1+M/\lambda_i^{new}} \right). \quad (14)$$

Note that, the right hand side is provided assuming the optimal LoRA-updated model $W_{lora}$ is also jointly diagonalizable with covariances $\Sigma$, $\Sigma^{new}$, and $W$.

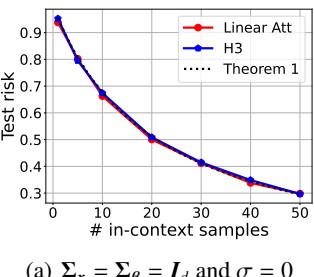
(a) $\Sigma_x = \Sigma_\beta = I_d$ and $\sigma = 0$

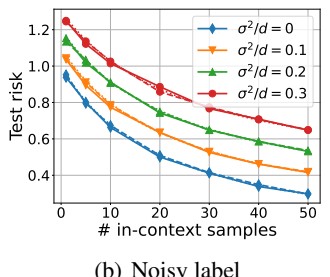
(b) Noisy label

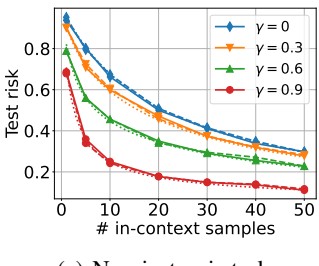
(c) Non-isotropic task

Figure 2: Empirical evidence validates Theorem 3.1 and Proposition 2.3. Experimental details are discussed in Section 4.

## 4. Experiments

We now conduct synthetic experiments to support our theoretical findings and further explore the behavior of different models of interest under different conditions. The experiments are designed to investigate various scenarios, including independent data, retrieval-augmented generation (RAG), task-feature alignment, low-rank parameterization, and LoRA adaption.

**Experimental setting.** We train 1-layer attention and H3 models for solving the linear regression ICL. As described in Section 2, we consider meta-learning setting where task parameter $\beta$ is randomly generated for each training sequence. In all experiments, we set the dimension $d = 20$. Depending on the in-context length ($n$), different models are trained to make in-context predictions. We train each model for 10000 iterations with batch size 128 and Adam optimizer with learning rate $10^{-3}$. Since our study focuses on the optimization landscape, and experiments are implemented via gradient descent, we repeat 20 model trainings from different initialization and results are presented as the minimal test risk among those 20 trails. In all the plots, theoretical predictions are obtained via the corresponding formulae presented in Section 3 and the test risks are normalized by the dimension $d$.

• **Equivalence among** $\mathcal{L}_{\text{PGD}}^\star$, $\mathcal{L}_{\text{ATT}}^\star$ **and** $\mathcal{L}_{\text{SSM}}^\star$ **(Figure 2).** To verify Proposition 2.3 as well as Theorem 3.1, we run random linear regression instances where in-context samples are generated obeying (7). Fig. 2(a) is identical to Fig. 1(a) where we set $\Sigma_x = \Sigma_\beta = I_d$ and $\sigma = 0$. In Fig. 2(b), set $\Sigma_x = \Sigma_\beta = I$ and vary noise level $\sigma^2$ from 0 to $0.3 \times d$. In Fig. 2(c), we consider noiseless labels, $\sigma = 0$, isotropic feature distribution $\Sigma_x = I_d$ and set task covariance to be $\Sigma_\beta = \gamma \mathbf{1}\mathbf{1}^\top + (1 - \gamma)I_d$ by choosing $\gamma$ in $\{0, 0.3, 0.6, 0.9\}$. Note that in Fig. 2(c), we train a sufficient number of models (greater than 20) to ensure the optimal model is obtained. In all the figures, solid and dashed curves correspond to the ICL results from training 1-layer ATT and SSM models, respectively, and dotted curves are obtained from (8) in Theorem 3.1. The alignment of solid, dashed and dotted curves validates our Proposition 2.3 and Theorem 3.1.

• **Distributional alignment experiments (Figs. 3(a)&3(b)).** In Figs. 3(a) and 3(b), we generate RAG and task-feature alignment data following (9) and (11), respectively, by setting $\sigma = 0$ and varying $\alpha$ from 0 to 0.6. Attention training results are displayed in solid curves, and we generate theory curve (dotted) via the $\mathcal{L}_\star$ formula as described in (36) in Appendix C.3 and (42) in Appendix C.4. The empirical alignments corroborate Theorems C.1 and C.2, further confirming that Proposition 2.3 is applicable to a broader range of real-world distributional alignment data.

• **Low-rank (Fig. 3(c)) and LoRA (Fig. 3(d)) experiments.** We also run simulations to verify our theoretical findings in Section 3.2. Consider the independent data setting as described in (7). In Fig. 3(c), we set $\Sigma_x = I_d$, $\sigma = 0$ and task covariance to be diagonal with diagonal entries $c[1 \; 2^{-1} \; \cdots \; d^{-1}]^\top$ for some normalization constant $c = d / \sum_{i=1}^d i^{-1}$, and parameterize the attention model using matrices $W_k, W_q \in \mathbb{R}^{(d+1) \times r}$ and vary $r$ across the set $\{1, 5, 10, 20\}$. Results show that empirical (solid) and theoretical (dotted, c.f. (13)) curves overlap. In Fig. 3(d), we implement two phases of training. *Phase 1:* Setting $\Sigma_x = \Sigma_\beta = I_d$ and $\sigma = 0$, we pretrain the model with full rank parameters and obtain weights $\hat{W}_k, \hat{W}_q, \hat{W}_v \in \mathbb{R}^{(d+1) \times (d+1)}$. *Phase 2:* We generate new examples with task covariance $\Sigma_\beta$ being a diagonal matrix with diagonal entries $c'[2^{-1} \; 2^{-2} \; \cdots \; 2^{-d}]^\top$ for some normalization constant $c' = d / \sum_{i=1}^d 2^{-i}$. Given the rank restriction $r$, we train additional LoRA parameters $W_{\text{up}}, W_{\text{down}} \in \mathbb{R}^{(d+1) \times r}$ where $W_{lora} := W_{\text{up}} W_{\text{down}}^\top$ and (2a) becomes $\text{ATT}(Z) = (Z(\hat{W}_q \hat{W}_k^\top + W_{\text{up}} W_{\text{down}}^\top)Z^\top)Z\hat{W}_v$. Fig. 3(d) presents the results after two phases of training where dotted curves are drawn from the right hand side of (14) directly. Here, note that since $\Sigma, \Sigma^{new}$ are diagonal, the right hand side of (14) returns the exact optimal risk of LoRA and the alignments verify it.

• **H3 outperforms linear attention (Figure 4).** Until now, our analysis has established the equivalence between linear attention and H3 models in solving linear ICL problem. Furthermore, we also investigate settings where H3 could outperform linear attention due to its sample weighting ability. In Figs. 4(a) and 4(b), instead of training separate models to fit the different context lengths, we train a single model with

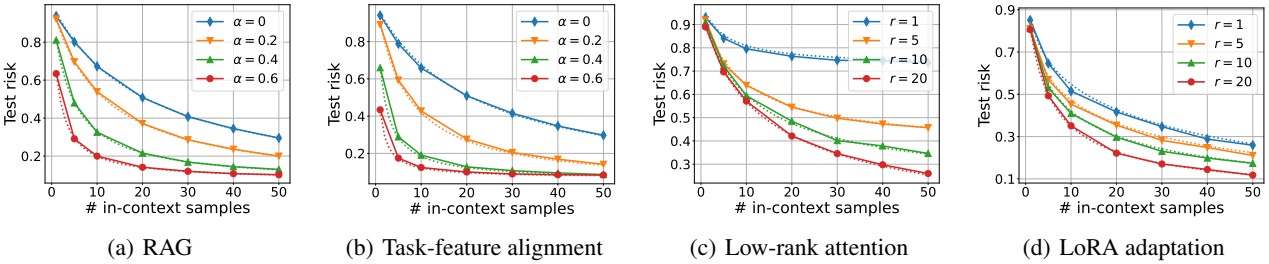

(a) RAG     (b) Task-feature alignment     (c) Low-rank attention     (d) LoRA adaptation

Figure 3: Distributional alignment and low-rank experiments. Experimental details are discussed in Section 4.

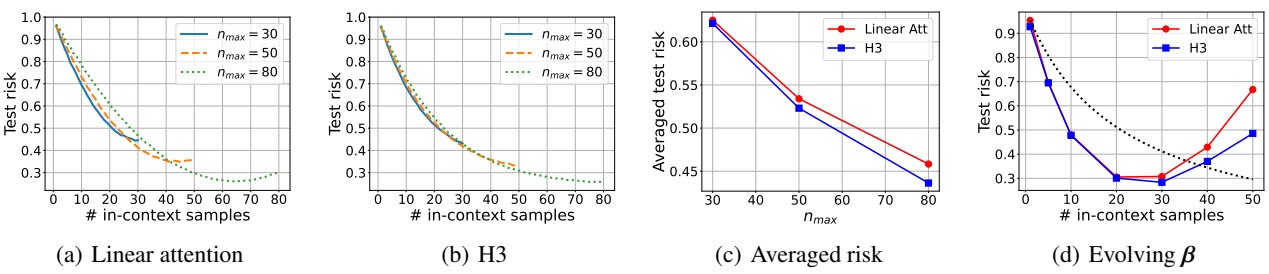

(a) Linear attention     (b) H3     (c) Averaged risk     (d) Evolving $\boldsymbol{\beta}$

Figure 4: Further comparison for linear attention and H3. Experimental details are discussed in Section 4.

fixed max-length $n_{\max}$ and loss is evaluated as the average loss given samples from 1 to $n_{\max}$. Such setting has been wildly studied in the previous ICL work (Garg et al., 2022; Akyürek et al., 2023; Li et al., 2023). We generate data according to (7) with $\boldsymbol{\Sigma_x} = \boldsymbol{\Sigma_\beta} = \boldsymbol{I_d}$ and $\sigma = 0$, and train 1-layer linear attention (Fig. 4(a)) and H3 (Fig. 4(b)) models with different max-lengths $n_{\max} = 30, 50, 80$. Comparison between Fig. 4(a) and 4(b) shows that 1-layer attention and H3 implement different algorithms in solving the averaged linear regression problem and H3 is more consistent in generalizing to longer context lengths. In Fig. 4(c), we plot the averaged risks for each model and H3 outperforms linear attention. Furthermore, in Fig. 4(d), we focus on the setting where in-context examples are generated using evolving task vector $\boldsymbol{\beta}$. Specifically, consider that each sequence corresponds to two individual task parameters $\boldsymbol{\beta}_1 \sim \mathcal{N}(0, \boldsymbol{I_d})$ and $\boldsymbol{\beta}_2 \sim \mathcal{N}(0, \boldsymbol{I_d})$. Then the $i$'th sample is generated via $\boldsymbol{x}_i \sim \mathcal{N}(0, \boldsymbol{I_d})$ and $y_i = \boldsymbol{\beta}_i^\top \boldsymbol{x}_i$ where $\boldsymbol{\beta}_i = \lambda_i \boldsymbol{\beta}_1 + (1 - \lambda_i)\boldsymbol{\beta}_2$ and $\lambda_i = i/n$. The results are reported in Fig. 4(d) which again shows that H3 achieves better performance compared to linear attention, as H3 may benefit from the additional convolutional filter (c.f. $\boldsymbol{f}$ in (2b)). Here, dotted curve represent the theoretical results under i.i.d. and noiseless setting, derived from Corollary 3.2.

## 5. Discussion

In this work, we revisited the loss landscape of in-context learning with 1-layer sequence models. We have established a general connection between ICL and gradient methods that accounts for correlated data, non-attention architectures (specifically SSMs), and the impact of low-rank parameter-

ization including LoRA adaptation. Our results elucidate two central findings: (1) The functions learned by different sequence model architectures exhibit a strong degree of *universality* and (ii) *Dataset and prompt design*, such as RAG, can substantially benefit ICL performance.

**Future directions and limitations.** The results of this work fall short of being a comprehensive theory for ICL in LLMs and can be augmented in multiple directions. First, while the exact equivalence between H3 and linear attention is remarkable, we should examine whether it extends to other SSMs. Secondly, while empirically predictive, our RAG and LoRA analyses are not precise and fully formal. Thirdly, it is desirable to develop a deeper understanding of multi-layer architectures and connect to iterative GD methods as in (Ahn et al., 2023; Von Oswald et al., 2023). Finally, we have studied the population risk of ICL training whereas one can also explore the sample complexity of pretraining (Wu et al., 2023; Lu et al., 2024). Moving beyond the theoretically tractable setup of this work, our simplified models are trained on in-context prompts from random initialization. Therefore, this theoretical study doesn't address more challenging in-context learning tasks, such as question answering, where both in-context demonstration and general knowledge from pretraining are required. Future work in this area could also shed light on how certain contexts might elicit undesirable behaviors acquired by an LLM during pretraining, an aspect not covered in our current analysis. This work also studies a theoretical model for retrieval augmentation-based ICL. In a real-life retrieval augmentation-based ICL, one needs to account for the quality of the collection of the retrievable demonstrations and its (negative) impacts on the final predictions.

## Acknowledgements

This work was supported in part by the National Science Foundation grants CCF-2046816, CCF-2403075, the Office of Naval Research award N000142412289, an Adobe Data Science Research award, and a gift by Google Research.

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

# A. Related Work

There is growing interest in understanding the mechanisms behind ICL (Brown et al., 2020; Liu et al., 2023b; Rae et al., 2021) in large language models (LLMs) due to its success in continuously enabling novel applications for LLMs (GeminiTeam et al., 2023; OpenAI, 2023; Touvron et al., 2023). Towards this, Xie et al. (2022) explain ICL by language model's ability to perform implicit Bayesian inference where, under specific assumptions on the pre-training data distribution, the model infers a shared latent concept among the in-context examples and leverages the concept to make a prediction. Müller et al. (2021); Hollmann et al. (2022); Müller et al. (2023) introduce prior-data fitted network (PFN) to approximate Bayesian inference on synthetic datasets and use it to perform downstream tasks such as tabular dataset classification. On the other hand, Olsson et al. (2022) posit induction heads as the key mechanism enabling ICL in Transformers. Park et al. (2024) study how various distributional properties of training data aid in the emergence of ICL in Transformers.

In the previous work, Garg et al. (2022) explored ICL ability of Transformers. In particular, they considered in-context prompts where each in-context example is labeled by a target function from a given function class, including linear models. A number of works have studied this and related settings to develop a theoretical understanding of ICL (von Oswald et al., 2023; Gatmiry et al.; Collins et al., 2024; Lin & Lee, 2024; Li et al., 2024; Bai et al., 2024; Akyürek et al., 2023; Zhang et al., 2023; Du et al., 2023). Akyürek et al. (2023) focus on linear regression and provide a construction of Transformer weights that can enable a single step of GD based on in-context examples. They further show that Transformers trained on in-context prompts exhibit behaviors similar to the models recovered via explicit learning algorithm on the in-context examples in a prompt. Along the similar line, Von Oswald et al. (2023) provide a construction of weights in linear attention-only Transformers that can emulate GD steps on in-context examples for a linear regression task. Interestingly, they find similarity between their constructed networks and the networks resulting from training on in-context prompts corresponding to linear regression tasks. Similar to this line of work, Dai et al. (2023) argue that pre-trained language models act as meta-optimizer which utilize attention to apply meta-gradients to the original language model based on the in-context examples. Focusing on various NLP tasks, they further connect it to a specific form of explicit fine-tuning that performs gradient updates to the attention-related parameters. Inspired by the connection between linear attention and GD, they developed a novel attention mechanism that mirrors the behavior of GD with momentum. Beyond Transformers, existing work (Lee et al., 2023; Zucchet et al., 2023; Grazzi et al., 2024) demonstrate that other model architectures, such as SSM and RNNs, are also capable of in-context learning (ICL).

Building on these primarily empirical studies, Zhang et al. (2024); Mahankali et al. (2024); Ahn et al. (2023); Duraisamy (2024) focus on developing a theoretical understanding of Transformers trained to perform ICL. For single-layer linear attention model trained on in-context prompts for random linear regression tasks with isotropic Gaussian features and isotropic Gaussian weight vectors, Mahankali et al. (2024); Ahn et al. (2023) show that the resulting model implements a single step of GD on in-context examples in a test prompt, thereby corroborating the findings of (Von Oswald et al., 2023). They also show that the learned model implements a PGD step, when faced with anisotropic Gaussian features, with Mahankali et al. (2024) also considering anisotropic Gaussian weight vectors. Ahn et al. (2023) further study multi-layer model and show that the trained model can implement a generalization of GD++ algorithm, supporting an empirical observation in (Von Oswald et al., 2023). On the other hand, Mahankali et al. (2024) extend their single-layer setup to consider suitable non-linear target functions, showing that learned Transformer again implements a single step of GD on lineare regression objective. For a single-layer linear attention model, Zhang et al. (2024) study the optimization dynamics of gradient flow while training such a model on in-context prompts for random linear regression tasks. Despite the non-convexity of the underlying problem, they show the convergence to the global minimum of the population objective. Similar to Mahankali et al. (2024); Ahn et al. (2023), they show that the trained model implements a single step of GD and PGD for isotropic and anisotropic Gaussian features, respectively. In addition, they also characterize the test-time prediction error for the trained model while highlighting its dependence on train and test prompt lengths. Interestingly, Zhang et al. (2024) further explore the effect of various distributional shifts, including the shift in task weight vector distributions between train and test time as well as the covariate shifts among train and test in-context prompts. Interestingly, they find that while linear-attention models are robust to most shifts, they exhibit brittleness to the covariate shifts.

While our work shares similarities with this line of works, as discussed in our contributions in the introduction, we expand the theoretical understanding of ICL along multiple novel dimensions, which includes the first study of LoRA adaptation for ICL in the presence of a distributional shift. Furthermore, we strive to capture the effect of retrieval augmentation (Lewis et al., 2020; Nakano et al., 2021) on ICL through our analysis. Retrieval augmentation allows for selecting most relevant demonstration out of a large collection for a test instance, e.g., via a dense retrieval model (Izacard et al., 2023), which can significantly outperform the typical ICL setup where fixed task-specific demonstrations are provided as in-context

examples (Wang et al., 2022; Basu et al., 2023). Through a careful modeling of retrieval augmentation via correlated design, we show that it indeed has a desirable amplification effect where the effective number in-context examples becomes larger with higher correlation which corresponds to preforming a successful retrieval of query-relevant demonstrations in a practical retrieval augmented setup.

Recently, state space models (SSMs) (Gu et al., 2021b;a; Fu et al., 2023; Gu & Dao, 2023) have appeared as potential alternatives to Transformer architecture, with more efficient scaling to input sequence length. Recent studies demonstrate that such SSMs can also perform ICL for simple non-language tasks (Park et al., 2024; Grazzi et al., 2024) as well as complex NLP tasks (Grazzi et al., 2024). That said, a rigorous theoretical understanding of ICL for SSMs akin to (Zhang et al., 2024; Mahankali et al., 2024; Ahn et al., 2023) is missing from the literature. In this work, we provide the first such theoretical treatment for ICL with SSMs. Focusing on H3 architecture (Fu et al., 2023), we highlight its advantages over linear attention in specific ICL settings.

## B. Equivalence among Gradient Descent, Attention, and State-Space Models

In this section, we present the proofs related to Section 2. Recap that given data

$$X = [x_1 \ \cdots \ x_n]^\top \in \mathbb{R}^{n \times d},$$
$$\xi = [\xi_1 \ \cdots \ \xi_n]^\top \in \mathbb{R}^n,$$
$$y = [y_1 \ \cdots \ y_n]^\top = X\beta + \xi \in \mathbb{R}^n,$$
$$Z_0 = [z_1 \ \ldots \ z_n \ \mathbf{0}_{d+1}]^\top = \begin{bmatrix} x_1 & \cdots & x_n & \mathbf{0}_d \\ y_1 & \cdots & y_n & 0 \end{bmatrix}^\top \in \mathbb{R}^{(n+1) \times (d+1)},$$

and corresponding prediction functions

$$g_{\text{PGD}}(Z) = x^\top W X^\top y, \tag{15a}$$
$$g_{\text{WPGD}}(Z) = x^\top W X^\top (\omega \odot y), \tag{15b}$$
$$g_{\text{ATT}}(Z) = (z^\top W_q W_k^\top Z_0^\top) Z_0 W_v v, \tag{15c}$$
$$g_{\text{SSM}}(Z) = \left( (z^\top W_q)^\top \odot ((Z_0 W_k \odot Z_0 W_v) * f)_{n+1} \right) v, \tag{15d}$$

we have objectives

$$\min_W \mathcal{L}_{\text{PGD}}(W) \quad \text{where} \quad \mathcal{L}_{\text{PGD}}(W) = \mathbb{E}\left[ (y - g_{\text{PGD}}(Z))^2 \right], \tag{16a}$$

$$\min_{W,\omega} \mathcal{L}_{\text{WPGD}}(W) \quad \text{where} \quad \mathcal{L}_{\text{WPGD}}(W) = \mathbb{E}\left[ (y - g_{\text{WPGD}}(Z))^2 \right], \tag{16b}$$

$$\min_{W_k, W_q, W_v, v} \mathcal{L}_{\text{ATT}}(W) \quad \text{where} \quad \mathcal{L}_{\text{ATT}}(W) = \mathbb{E}\left[ (y - g_{\text{ATT}}(Z))^2 \right], \tag{16c}$$

$$\min_{W_k, W_q, W_v, v, f} \mathcal{L}_{\text{SSM}}(W) \quad \text{where} \quad \mathcal{L}_{\text{SSM}}(W) = \mathbb{E}\left[ (y - g_{\text{SSM}}(Z))^2 \right]. \tag{16d}$$

Here, the expectation is over the randomness in $(x_i, \xi_i)_{i=1}^n$ and $\beta$, and the search space for $W$ is $\mathbb{R}^{d \times d}$, for $\omega$ is $\mathbb{R}^n$, for $W_k, W_q, W_v$ is $\mathbb{R}^{(d+1) \times (d+1)}$, for $v$ is $\mathbb{R}^{d+1}$, and for $f$ is $\mathbb{R}^{n+1}$.

### B.1. Proof of Proposition 2.3

Consider the problem setting as discussed in Section 2, Proposition 2.3 can be proven by the following two lemmas.

**Lemma B.1.** *Suppose Assumptions 2.1 and 2.2 hold. Then, given the objectives* (16a) *and* (16c)*, we have*

$$\min_{W_q, W_k, W_v, v} \mathcal{L}_{ATT}(W) = \min_W \mathcal{L}_{PGD}(W).$$

*Proof.* Recap the linear attention estimator from (15c) and denote

$$W_q W_k^\top = \begin{bmatrix} \bar{W} & w_1 \\ w_2^\top & w \end{bmatrix} \quad \text{and} \quad W_v v = \begin{bmatrix} v_1 \\ v \end{bmatrix},$$

where $\bar{W} \in \mathbb{R}^{d \times d}$, $w_1, w_2, v_1 \in \mathbb{R}^d$, and $w, v \in \mathbb{R}$. Then we have

$$
\begin{aligned}
g_{\text{ATT}}(Z) &= (z^\top W_q W_k^\top Z_0^\top) Z_0 W_v v \\
&= [x^\top \ 0] \begin{bmatrix} \bar{W} & w_1 \\ w_2^\top & w \end{bmatrix} \begin{bmatrix} X^\top & 0_d \\ y^\top & 0 \end{bmatrix} \begin{bmatrix} X & y \\ 0_d^\top & 0 \end{bmatrix} \begin{bmatrix} v_1 \\ v \end{bmatrix} \\
&= (x^\top \bar{W} X^\top + x^\top w_1 y^\top)(X v_1 + yv) \\
&= x^\top (v\bar{W}) X^\top y + x^\top w_1 y^\top X v_1 + x^\top \left( \bar{W} X^\top X v_1 + v \|y\|_{\ell_2}^2 w_1 \right) \\
&= x^\top (v\bar{W} + w_1 v_1^\top) X^\top y + x^\top \left( \bar{W} X^\top X v_1 + v \|y\|_{\ell_2}^2 w_1 \right) \\
&= \underbrace{x^\top \tilde{W} X^\top y}_{\tilde{g}_{\text{ATT}}(Z)} + \underbrace{x^\top \left( \bar{W} X^\top X v_1 + v \|y\|_{\ell_2}^2 w_1 \right)}_{\varepsilon},
\end{aligned}
\tag{17}
$$

where $\tilde{W} := v\bar{W} + w_1 v_1^\top$.

We first show that for any given parameters $W_k, W_q, W_v, v$,

$$
\mathbb{E}\left[ (g_{\text{ATT}}(Z) - y)^2 \right] \geq \mathbb{E}\left[ (\tilde{g}_{\text{ATT}}(Z) - y)^2 \right].
\tag{18}
$$

To this goal, we have

$$
\begin{aligned}
\mathbb{E}\left[ (g_{\text{ATT}}(Z) - y)^2 \right] - \mathbb{E}\left[ (\tilde{g}_{\text{ATT}}(Z) - y)^2 \right] &= \mathbb{E}\left[ (\tilde{g}_{\text{ATT}}(Z) + \varepsilon - y)^2 \right] - \mathbb{E}\left[ (\tilde{g}_{\text{ATT}}(Z) - y)^2 \right] \\
&= \mathbb{E}[\varepsilon^2] + 2\,\mathbb{E}[(\tilde{g}_{\text{ATT}}(Z) - y)\varepsilon]
\end{aligned}
\tag{19}
$$

where we have decomposition

$$
\begin{aligned}
(\tilde{g}_{\text{ATT}}(Z) - y)\varepsilon &= (x^\top \tilde{W} X^\top y - y) x^\top \left( \bar{W} X^\top X v_1 + v \|y\|_{\ell_2}^2 w_1 \right) \\
&= y^\top X \tilde{W}^\top x x^\top \left( \bar{W} X^\top X v_1 + v \|y\|_{\ell_2}^2 w_1 \right) - y x^\top \left( \bar{W} X^\top X v_1 + v \|y\|_{\ell_2}^2 w_1 \right) \\
&= \underbrace{y^\top X \tilde{W}^\top x x^\top \bar{W} X^\top X v_1}_{(a)} + \underbrace{v \|y\|_{\ell_2}^2 y^\top X \tilde{W}^\top x x^\top w_1}_{(b)} - \underbrace{y x^\top \bar{W} X^\top X v_1}_{(c)} - \underbrace{vy \|y\|_{\ell_2}^2 x^\top w_1}_{(d)}.
\end{aligned}
$$

In the following, we consider the expectations of $(a), (b), (c), (d)$ sequentially, which return zeros under Assumptions 2.1 and 2.2. Note that since Assumption 2.1 holds, expectation of any odd *order* of monomial of the entries of $X, x, \beta$ returns zero, i.e., order of $x^\top \beta x$ is 3 and therefore $\mathbb{E}[x^\top \beta x] = 0_d$.

$$
\begin{aligned}
(a): \quad & \mathbb{E}\left[ y^\top X \tilde{W}^\top x x^\top \bar{W} X^\top X v_1 \right] \\
&= \mathbb{E}\left[ (X\beta + \xi)^\top X \tilde{W}^\top x x^\top \bar{W} X^\top X v_1 \right] \\
&= \mathbb{E}\left[ \beta^\top X^\top X \tilde{W}^\top x x^\top \bar{W} X^\top X v_1 \right] + \mathbb{E}\left[ \xi^\top X \tilde{W}^\top x x^\top \bar{W} X^\top X v_1 \right] \\
&= 0.
\end{aligned}
$$

$$
\begin{aligned}
(b): \quad & \mathbb{E}\left[ v \|y\|_{\ell_2}^2 y^\top X \tilde{W}^\top x x^\top w_1 \right] \\
&= \mathbb{E}\left[ v(X\beta + \xi)^\top (X\beta + \xi)(X\beta + \xi)^\top X \tilde{W}^\top x x^\top w_1 \right] \\
&= \mathbb{E}\left[ v \|\xi\|_{\ell_2}^2 \xi^\top X \tilde{W}^\top x x^\top w_1 \right] \\
&= 0.
\end{aligned}
$$

$$
\begin{aligned}
(c): \quad & \mathbb{E}\left[ y x^\top \bar{W} X^\top X v_1 \right] \\
&= \mathbb{E}\left[ (x^\top \beta + \xi) x^\top \bar{W} X^\top X v_1 \right] \\
&= \mathbb{E}\left[ \beta^\top x x^\top \bar{W} X^\top X v_1 \right] + \mathbb{E}\left[ \xi x^\top \bar{W} X^\top X v_1 \right] \\
&= 0.
\end{aligned}
$$

$$(d): \quad \mathbb{E}\left[vy \|\boldsymbol{y}\|_{\ell_2}^2 \boldsymbol{x}^\top \boldsymbol{w}_1\right]$$
$$= v\mathbb{E}\left[(\boldsymbol{\beta}^\top \boldsymbol{x} + \xi)(\boldsymbol{X}\boldsymbol{\beta} + \boldsymbol{\xi})^\top (\boldsymbol{X}\boldsymbol{\beta} + \boldsymbol{\xi})\boldsymbol{x}^\top \boldsymbol{w}_1\right]$$
$$= v\mathbb{E}\left[\xi \|\boldsymbol{\xi}\|_{\ell_2}^2 \boldsymbol{x}^\top \boldsymbol{w}_1\right]$$
$$= 0.$$

Combining the results with (19) returns that

$$\mathbb{E}\left[(g_{\text{ATT}}(\boldsymbol{Z}) - y)^2\right] - \mathbb{E}\left[(\tilde{g}_{\text{ATT}}(\boldsymbol{Z}) - y)^2\right] = \mathbb{E}[\varepsilon^2] \geq 0 \tag{20}$$

which completes the proof of (18). Therefore, we obtain

$$\min_{\boldsymbol{W}_q, \boldsymbol{W}_k, \boldsymbol{W}_v, \boldsymbol{v}} \mathbb{E}\left[(g_{\text{ATT}}(\boldsymbol{Z}) - y)^2\right] \geq \min_{\tilde{\boldsymbol{W}}} \mathbb{E}\left[(\tilde{g}_{\text{ATT}}(\boldsymbol{Z}) - y)^2\right] = \min_{\boldsymbol{W}} \mathbb{E}\left[(g_{\text{PGD}}(\boldsymbol{Z}) - y)^2\right].$$

We conclude the proof of this lemma by showing that for any $\boldsymbol{W} \in \mathbb{R}^{d \times d}$ in $g_{\text{PGD}}$, there exist $\boldsymbol{W}_k, \boldsymbol{W}_q, \boldsymbol{W}_v, \boldsymbol{v}$ such that $g_{\text{ATT}}(\boldsymbol{Z}) = g_{\text{PGD}}(\boldsymbol{Z})$. Let

$$\boldsymbol{W}_k = \boldsymbol{W}_v = \boldsymbol{I}_{d+1}, \qquad \boldsymbol{W}_q = \begin{bmatrix} \boldsymbol{W} & \boldsymbol{0}_d \\ \boldsymbol{0}_d^\top & 0 \end{bmatrix}, \quad \text{and} \quad \boldsymbol{v} = \begin{bmatrix} \boldsymbol{0}_d \\ 1 \end{bmatrix}.$$

Then we obtain

$$g_{\text{ATT}}(\boldsymbol{Z}) = \boldsymbol{x}^\top \boldsymbol{W} \boldsymbol{X}^\top \boldsymbol{y} = g_{\text{PGD}}(\boldsymbol{Z}), \tag{21}$$

which completes the proof. □

**Lemma B.2.** *Suppose Assumptions 2.1 and 2.2 hold. Then, given the objectives in (16), we have*

$$\min_{\boldsymbol{W}_q, \boldsymbol{W}_k, \boldsymbol{W}_v, \boldsymbol{v}, \boldsymbol{f}} \mathcal{L}_{\text{SSM}}(\mathcal{W}) = \min_{\boldsymbol{W}, \omega} \mathcal{L}_{\text{WPGD}}(\mathcal{W}). \tag{22}$$

*Additionally, if the examples $(\boldsymbol{x}_i, y_i)_{i=1}^n$ follow the same distribution and are conditionally independent given $\boldsymbol{x}$ and $\boldsymbol{\beta}$, then SSM/H3 can achieve the optimal loss using the all-ones filter and*

$$\min_{\boldsymbol{W}, \omega} \mathcal{L}_{\text{WPGD}}(\mathcal{W}) = \min_{\boldsymbol{W}} \mathcal{L}_{\text{PGD}}(\mathcal{W}). \tag{23}$$

*Proof.* Recap the SSM estimator from (15d) and let

$$\boldsymbol{W}_q = \begin{bmatrix} \boldsymbol{w}_{q1} & \boldsymbol{w}_{q2} & \cdots & \boldsymbol{w}_{q,d+1} \end{bmatrix},$$
$$\boldsymbol{W}_k = \begin{bmatrix} \boldsymbol{w}_{k1} & \boldsymbol{w}_{k2} & \cdots & \boldsymbol{w}_{k,d+1} \end{bmatrix},$$
$$\boldsymbol{W}_v = \begin{bmatrix} \boldsymbol{w}_{v1} & \boldsymbol{w}_{v2} & \cdots & \boldsymbol{w}_{v,d+1} \end{bmatrix},$$

where $\boldsymbol{w}_{qj}, \boldsymbol{w}_{kj}, \boldsymbol{w}_{vj} \in \mathbb{R}^{d+1}$ for $j \leq d + 1$, and let

$$\boldsymbol{v} = \begin{bmatrix} v_1 \\ v_2 \\ \cdots \\ v_{d+1} \end{bmatrix}, \quad \text{and} \quad \boldsymbol{f} = \begin{bmatrix} f_0 \\ f_1 \\ \cdots \\ f_n \end{bmatrix}.$$

Then we have

$$g_{\text{SSM}}(\boldsymbol{Z}) = \left((\boldsymbol{z}^\top \boldsymbol{W}_q)^\top \odot ((\boldsymbol{Z}_0 \boldsymbol{W}_k \odot \boldsymbol{Z}_0 \boldsymbol{W}_v) * \boldsymbol{f})_{n+1}\right) \boldsymbol{v}$$

$$= \sum_{i=1}^n f_{n+1-i} \cdot \boldsymbol{v}^\top \left(\begin{bmatrix} \boldsymbol{w}_{q1}^\top \boldsymbol{z} \\ \cdots \\ \boldsymbol{w}_{q,d+1}^\top \boldsymbol{z} \end{bmatrix} \odot \begin{bmatrix} \boldsymbol{w}_{k1}^\top \boldsymbol{z}_i \boldsymbol{w}_{v1}^\top \boldsymbol{z}_i \\ \cdots \\ \boldsymbol{w}_{k,d+1}^\top \boldsymbol{z}_i \boldsymbol{w}_{v,d+1}^\top \boldsymbol{z}_i \end{bmatrix}\right)$$

$$= \sum_{i=1}^n f_{n+1-i} \cdot \boldsymbol{v}^\top \begin{bmatrix} \boldsymbol{w}_{q1}^\top \boldsymbol{z} \boldsymbol{w}_{k1}^\top \boldsymbol{z}_i \boldsymbol{w}_{v1}^\top \boldsymbol{z}_i \\ \cdots \\ \boldsymbol{w}_{q,d+1}^\top \boldsymbol{z} \boldsymbol{w}_{k,d+1}^\top \boldsymbol{z}_i \boldsymbol{w}_{v,d+1}^\top \boldsymbol{z}_i \end{bmatrix}.$$

Next for all $j \leq d + 1$, let

$$\boldsymbol{w}_{qj} = \begin{bmatrix} \bar{\boldsymbol{w}}_{qj} \\ w_{qj} \end{bmatrix}, \quad \boldsymbol{w}_{kj} = \begin{bmatrix} \bar{\boldsymbol{w}}_{kj} \\ w_{kj} \end{bmatrix}, \quad \boldsymbol{w}_{vj} = \begin{bmatrix} \bar{\boldsymbol{w}}_{vj} \\ w_{vj} \end{bmatrix}$$

where $\bar{\boldsymbol{w}}_{qj}, \bar{\boldsymbol{w}}_{kj}, \bar{\boldsymbol{w}}_{vj} \in \mathbb{R}^d$ and $w_{qj}, w_{kj}, w_{vj} \in \mathbb{R}$. Then we have

$$\begin{aligned}
\boldsymbol{w}_{qj}^\top \boldsymbol{z} \boldsymbol{w}_{kj}^\top \boldsymbol{z}_i \boldsymbol{w}_{vj}^\top \boldsymbol{z}_i &= \left( \bar{\boldsymbol{w}}_{qj}^\top \boldsymbol{x} \right) \left( \bar{\boldsymbol{w}}_{kj}^\top \boldsymbol{x}_i + w_{kj} y_i \right) \left( \bar{\boldsymbol{w}}_{vj}^\top \boldsymbol{x}_i + w_{vj} y_i \right) \\
&= \boldsymbol{x}^\top \bar{\boldsymbol{w}}_{qj} \left( w_{vj} \bar{\boldsymbol{w}}_{kj}^\top + w_{kj} \bar{\boldsymbol{w}}_{vj}^\top \right) \boldsymbol{x}_i y_i + \left( \bar{\boldsymbol{w}}_{qj}^\top \boldsymbol{x} \right) \left( \bar{\boldsymbol{w}}_{kj}^\top \boldsymbol{x}_i \right) \left( \bar{\boldsymbol{w}}_{vj}^\top \boldsymbol{x}_i \right) + \left( w_{kj} w_{vj} \bar{\boldsymbol{w}}_{qj}^\top \boldsymbol{x} y_i^2 \right) \\
&= \boldsymbol{x}^\top \boldsymbol{W}_j' \boldsymbol{x}_i y_i + \delta_j(\boldsymbol{x}, \boldsymbol{x}_i, \boldsymbol{x}_i) + {\boldsymbol{w}_j'}^\top \boldsymbol{x} y_i^2
\end{aligned}$$

where

$$\begin{aligned}
\boldsymbol{W}_j' &:= \bar{\boldsymbol{w}}_{qj} \left( w_{vj} \bar{\boldsymbol{w}}_{kj}^\top + w_{kj} \bar{\boldsymbol{w}}_{vj}^\top \right) \in \mathbb{R}^{d \times d}, \\
\boldsymbol{w}_j' &:= w_{kj} w_{vj} \bar{\boldsymbol{w}}_{qj} \in \mathbb{R}^d, \\
\delta_j(\boldsymbol{x}, \boldsymbol{x}_i, \boldsymbol{x}_i) &:= \left( \bar{\boldsymbol{w}}_{qj}^\top \boldsymbol{x} \right) \left( \bar{\boldsymbol{w}}_{kj}^\top \boldsymbol{x}_i \right) \left( \bar{\boldsymbol{w}}_{vj}^\top \boldsymbol{x}_i \right) \in \mathbb{R}.
\end{aligned}$$

Then

$$\begin{aligned}
g_{\text{SSM}}(\boldsymbol{Z}) &= \sum_{i=1}^n f_{n+1-i} \cdot \sum_{j=1}^{d+1} v_j \left( \boldsymbol{x}^\top \boldsymbol{W}_j' \boldsymbol{x}_i y_i + \delta_j(\boldsymbol{x}, \boldsymbol{x}_i, \boldsymbol{x}_i) + {\boldsymbol{w}_j'}^\top \boldsymbol{x} y_i^2 \right) \\
&= \boldsymbol{x}^\top \left( \sum_{j=1}^{d+1} v_j \boldsymbol{W}_j' \right) \boldsymbol{X} (\boldsymbol{y} \odot \tilde{\boldsymbol{f}}) + \sum_{i=1}^n f_{n+1-i} \cdot \sum_{j=1}^{d+1} v_j \cdot \delta_j(\boldsymbol{x}, \boldsymbol{x}_i, \boldsymbol{x}_i) + \left( \sum_{j=1}^{d+1} v_j {\boldsymbol{w}_j'}^\top \right) \boldsymbol{x} \boldsymbol{y}^\top (\boldsymbol{y} \odot \tilde{\boldsymbol{f}}) \\
&= \underbrace{\boldsymbol{x}^\top \tilde{\boldsymbol{W}} \boldsymbol{X} \tilde{\boldsymbol{y}}}_{\tilde{g}_{\text{SSM}}(\boldsymbol{Z})} + \underbrace{\tilde{\delta}(\boldsymbol{x}, \boldsymbol{X}, \boldsymbol{X})}_{\varepsilon_1} + \underbrace{\tilde{\boldsymbol{w}}^\top \boldsymbol{x} \boldsymbol{y}^\top \tilde{\boldsymbol{y}}}_{\varepsilon_2}.
\end{aligned}$$

where

$$\begin{aligned}
\tilde{\boldsymbol{f}} &:= [f_n \; \cdots \; f_1]^\top \in \mathbb{R}^n, \\
\tilde{\boldsymbol{y}} &:= \boldsymbol{y} \odot \tilde{\boldsymbol{f}} \in \mathbb{R}^n, \\
\tilde{\boldsymbol{W}} &:= \sum_{j=1}^{d+1} v_j \boldsymbol{W}_j' \in \mathbb{R}^{d \times d}, \\
\tilde{\boldsymbol{w}} &:= \sum_{j=1}^{d+1} v_j \boldsymbol{w}_j' \in \mathbb{R}^d, \\
\tilde{\delta}(\boldsymbol{x}, \boldsymbol{X}, \boldsymbol{X}) &:= \sum_{i=1}^n f_{n+1-i} \cdot \sum_{j=1}^{d+1} v_j \cdot \delta_j(\boldsymbol{x}, \boldsymbol{x}_i, \boldsymbol{x}_i) \in \mathbb{R}.
\end{aligned}$$

Next we will show that for any $\boldsymbol{W}_k, \boldsymbol{W}_q, \boldsymbol{W}_v, \boldsymbol{v}$,

$$\mathbb{E}\left[ (g_{\text{SSM}}(\boldsymbol{Z}) - y)^2 \right] \geq \mathbb{E}\left[ (\tilde{g}_{\text{SSM}}(\boldsymbol{Z}) - y)^2 \right].$$

To start with, we obtain

$$\begin{aligned}
\mathbb{E}\left[ (g_{\text{SSM}}(\boldsymbol{Z}) - y)^2 \right] &= \mathbb{E}\left[ (\tilde{g}_{\text{SSM}}(\boldsymbol{Z}) + \varepsilon_1 + \varepsilon_2 - y)^2 \right] \\
&= \mathbb{E}\left[ (\tilde{g}_{\text{SSM}}(\boldsymbol{Z}) - y)^2 \right] + \mathbb{E}\left[ (\varepsilon_1 + \varepsilon_2)^2 \right] + 2 \mathbb{E}\left[ (\tilde{g}_{\text{SSM}}(\boldsymbol{Z}) - y)(\varepsilon_1 + \varepsilon_2) \right]
\end{aligned} \tag{24}$$

where there is decomposition

$$(\tilde{g}_{\text{SSM}}(\boldsymbol{Z}) - y)(\varepsilon_1 + \varepsilon_2) = \underbrace{\tilde{\delta}(\boldsymbol{x}, \boldsymbol{X}, \boldsymbol{X}) \cdot \boldsymbol{x}^\top \tilde{\boldsymbol{W}} \boldsymbol{X} \tilde{\boldsymbol{y}}}_{(a)} - \underbrace{\tilde{\delta}(\boldsymbol{x}, \boldsymbol{X}, \boldsymbol{X}) y}_{(b)} + \underbrace{\tilde{\boldsymbol{w}}^\top \boldsymbol{x} \boldsymbol{y}^\top \tilde{\boldsymbol{y}} \cdot \boldsymbol{x}^\top \tilde{\boldsymbol{W}} \boldsymbol{X} \tilde{\boldsymbol{y}}}_{(c)} - \underbrace{y \cdot \tilde{\boldsymbol{w}}^\top \boldsymbol{x} \boldsymbol{y}^\top \tilde{\boldsymbol{y}}}_{(d)}.$$

In the following, similar to the proof of Lemma B.1, we consider the expectations of $(a), (b), (c), (d)$ sequentially, which return zeros under Assumptions 2.1 and 2.2. Note that $\delta_j(x, x_i, x_i)$'s and $\tilde{\delta}(x, X, X)$ are summation of monomials of entries of $(x, X, \beta)$ with order 3, and entries of $y$ and $y$ are summation of monomials of entries of $(x, X, \beta)$ with even orders: e.g., $y = x^\top \beta + \xi$ where $\xi$ is of oder 0 and $x^\top \beta$ is of order 2.

$$
\begin{aligned}
(a): \quad & \mathbb{E}\left[\tilde{\delta}(x, X, X) \cdot x^\top \tilde{W} X \tilde{y}\right] \\
&= \mathbb{E}\left[\tilde{\delta}(x, X, X) \cdot x^\top \tilde{W} X(X\beta \odot \tilde{f})\right] + \mathbb{E}\left[\tilde{\delta}(x, X, X) \cdot x^\top \tilde{W} X(\xi \odot \tilde{f})\right] \\
&= \mathbb{E}\left[\tilde{\delta}(x, X, X) \cdot x^\top \tilde{W} X\right]\mathbb{E}\left[\xi \odot \tilde{f}\right] \\
&= 0.
\end{aligned}
$$

$$
\begin{aligned}
(b): \quad & \mathbb{E}\left[\tilde{\delta}(x, X, X)y\right] \\
&= \mathbb{E}\left[\tilde{\delta}(x, X, X)(x^\top \beta + \xi)\right] \\
&= \mathbb{E}\left[\tilde{\delta}(x, X, X)x^\top \beta\right] + \mathbb{E}\left[\tilde{\delta}(x, X, X)\xi\right] \\
&= 0.
\end{aligned}
$$

$$
\begin{aligned}
(c): \quad & \mathbb{E}\left[\tilde{w}^\top x y^\top \tilde{y} \cdot x^\top \tilde{W} X \tilde{y}\right] \\
&= \mathbb{E}\left[\tilde{w}^\top x(X\beta + \xi)^\top(X\beta \odot \tilde{f} + \xi \odot \tilde{f}) \cdot x^\top \tilde{W} X(X\beta \odot \tilde{f} + \xi \odot \tilde{f})\right] \\
&= 0.
\end{aligned}
$$

$$
\begin{aligned}
(d): \quad & \mathbb{E}\left[y \cdot \tilde{w}^\top x y^\top \tilde{y}\right] \\
&= \mathbb{E}\left[(x^\top \beta + \xi) \cdot \tilde{w}^\top x(X\beta + \xi)^\top(X\beta \odot \tilde{f} + \xi \odot \tilde{f})\right] \\
&= 0.
\end{aligned}
$$

Combining the results with (24) results that

$$
\mathbb{E}\left[(g_{\text{SSM}}(Z) - y)^2\right] - \mathbb{E}\left[(\tilde{g}_{\text{SSM}}(Z) - y)^2\right] = \mathbb{E}\left[(\varepsilon_1 + \varepsilon_2)^2\right] \geq 0.
$$

Therefore we obtain,

$$
\min_{W_q, W_k, W_v, v, f} \mathbb{E}\left[(g_{\text{SSM}}(Z) - y)^2\right] \geq \min_{\tilde{W}, \tilde{f}} \mathbb{E}\left[(\tilde{g}_{\text{SSM}}(Z) - y)^2\right] = \min_{W, \omega} \mathbb{E}\left[(g_{\text{WPGD}}(Z) - y)^2\right].
$$

Next we show that for any choices of $W$ and $\omega$ in $g_{\text{WPGD}}$, there are $W_{q,k,v}, v, f$ such that $g_{\text{SSM}} \equiv g_{\text{WPGD}}$. To this end, given $\omega = [\omega_1 \; \ldots \; \omega_n]^\top$, let

$$
W_q = I_{d+1}, \quad W_k = \begin{bmatrix} W^\top & 0_d \\ 0_d^\top & 0 \end{bmatrix}, \quad W_v = \begin{bmatrix} 0_{d\times d} & 0_d \\ 1_d^\top & 0 \end{bmatrix}, \quad v = \begin{bmatrix} 1_d \\ 0 \end{bmatrix} \quad \text{and} \quad f = \begin{bmatrix} 0 \\ \omega_n \\ \ldots \\ \omega_1 \end{bmatrix}.
$$

Then we get

$$
\begin{aligned}
((Z_0 W_k \odot Z_0 W_v) * f)_{n+1} &= \left(\left(\begin{bmatrix} XW^\top & 0_n \\ 0_d & 0 \end{bmatrix} \odot \begin{bmatrix} y1_d^\top & 0_n \\ 0_d & 0 \end{bmatrix}\right) * f\right)_{n+1} \\
&= \begin{bmatrix} \sum_{i=1}^n \omega_i \cdot y_i W x_i \\ 0 \end{bmatrix} \\
&= \begin{bmatrix} WX^\top(y \odot \omega) \\ 0 \end{bmatrix},
\end{aligned}
$$

and therefore

$$g_{\text{SSM}}(\boldsymbol{Z}) = \boldsymbol{x}^\top \boldsymbol{W} \boldsymbol{X}^\top (\boldsymbol{y} \odot \boldsymbol{\omega}) = g_{\text{WPGD}}(\boldsymbol{Z}),$$

which completes the proof of (22).

Next, to show (23), for any $\boldsymbol{W} \in \mathbb{R}^{d \times d}$, let $\mathcal{L}(\boldsymbol{\omega}) = \mathbb{E}\left[(\boldsymbol{x}^\top \boldsymbol{W} \boldsymbol{X}^\top (\boldsymbol{y} \odot \boldsymbol{\omega}) - y)^2\right]$. Then we have

$$
\begin{aligned}
\frac{\partial \mathcal{L}(\boldsymbol{\omega})}{\partial \omega_i} &= \mathbb{E}\left[2\left(\boldsymbol{x}^\top \boldsymbol{W} \sum_{j=1}^n \omega_j y_j \boldsymbol{x}_j - y\right)(\boldsymbol{x}^\top \boldsymbol{W} y_i \boldsymbol{x}_i)\right] \\
&= 2 \sum_{j=1}^n \omega_j \mathbb{E}\left[(\boldsymbol{x}^\top \boldsymbol{W} y_j \boldsymbol{x}_j)(\boldsymbol{x}^\top \boldsymbol{W} y_i \boldsymbol{x}_i)\right] - 2 \mathbb{E}\left[y \boldsymbol{x}^\top \boldsymbol{W} y_i \boldsymbol{x}_i\right].
\end{aligned}
$$

Here since $(\boldsymbol{x}_i, y_i)_{i=1}^n$ follow the same distribution and are conditionally independent given $\boldsymbol{x}$ and $\boldsymbol{\beta}$, for any $i \neq j \neq j'$, $\mathbb{E}\left[(\boldsymbol{x}^\top \boldsymbol{W} y_i \boldsymbol{x}_i)^2\right] = \mathbb{E}\left[(\boldsymbol{x}^\top \boldsymbol{W} y_j \boldsymbol{x}_j)^2\right]$ and $\mathbb{E}\left[(\boldsymbol{x}^\top \boldsymbol{W} y_j \boldsymbol{x}_j)(\boldsymbol{x}^\top \boldsymbol{W} y_i \boldsymbol{x}_i)\right] = \mathbb{E}\left[(\boldsymbol{x}^\top \boldsymbol{W} y_{j'} \boldsymbol{x}_{j'})(\boldsymbol{x}^\top \boldsymbol{W} y_i \boldsymbol{x}_i)\right]$. Then let

$$\mathbb{E}\left[(\boldsymbol{x}^\top \boldsymbol{W} y_j \boldsymbol{x}_j)(\boldsymbol{x}^\top \boldsymbol{W} y_i \boldsymbol{x}_i)\right] = \begin{cases} c_1, & i \neq j \\ c_2, & i = j \end{cases} \quad \text{and} \quad \mathbb{E}\left[y \boldsymbol{x}^\top \boldsymbol{W} y_i \boldsymbol{x}_i\right] = c_3,$$

where $(c_1, c_2, c_3) := (c_1(\boldsymbol{W}), c_2(\boldsymbol{W}), c_3(\boldsymbol{W}))$. We get

$$\frac{\partial \mathcal{L}(\boldsymbol{\omega})}{\partial \omega_i} = 2c_1 \boldsymbol{\omega}^\top \mathbf{1}_n + 2(c_2 - c_1)\omega_i - 2c_3.$$

If $c_2 - c_1 = 0$, then $\frac{\partial \mathcal{L}(\boldsymbol{\omega})}{\partial \omega_i} \equiv 2c_1 \boldsymbol{\omega}^\top \mathbf{1}_n - 2c_3$ for all $i \leq n$ and any $\boldsymbol{\omega} \in \mathbb{R}^n$ achieves the same performance.

If $c_2 - c_1 \neq 0$, setting $\frac{\partial \mathcal{L}(\boldsymbol{\omega})}{\partial \omega_i} = 0$ returns

$$\omega_i = \frac{c_3 - c_1 \sum_{j=1}^n \omega_j}{c_2 - c_1} := C \quad \text{for all } i \leq n.$$

Therefore the optimal loss is achieved via setting $\boldsymbol{\omega} = C \mathbf{1}_n$. Without loss of generality, we can update $\boldsymbol{W} \to C\boldsymbol{W}$. Then $\boldsymbol{\omega} = \mathbf{1}_n$, and we obtain

$$\min_{\boldsymbol{W}, \boldsymbol{\omega}} \mathbb{E}\left[\left(\boldsymbol{x}^\top \boldsymbol{W} \boldsymbol{X}^\top (\boldsymbol{y} \odot \boldsymbol{\omega}) - y\right)^2\right] = \min_{\boldsymbol{W}} \mathbb{E}\left[(\boldsymbol{x}^\top \boldsymbol{W} \boldsymbol{X}^\top \boldsymbol{y} - y)^2\right]$$

which completes the proof of (23). □

### B.2. Proof of Lemma 2.4

*Proof.* Recap the loss $\mathcal{L}_{\text{PGD}}(\boldsymbol{W})$ in (16a) and prediction $g_{\text{PGD}}(\boldsymbol{Z})$ in (15a), we have

$$
\begin{aligned}
\mathcal{L}_{\text{PGD}}(\boldsymbol{W}) &= \mathbb{E}[(y - g_{\text{PGD}}(\boldsymbol{Z}))^2] \\
&= \mathbb{E}\left[\left(\boldsymbol{x}^\top \boldsymbol{\beta} + \xi - \boldsymbol{x}^\top \boldsymbol{W} \boldsymbol{X}^\top (\boldsymbol{X}\boldsymbol{\beta} + \boldsymbol{\xi})\right)^2\right] \\
&= \mathbb{E}\left[(\boldsymbol{x}^\top \boldsymbol{\beta} - \boldsymbol{x}^\top \boldsymbol{W} \boldsymbol{X}^\top \boldsymbol{X}\boldsymbol{\beta})^2 + 2(\boldsymbol{x}^\top \boldsymbol{\beta} - \boldsymbol{x}^\top \boldsymbol{W} \boldsymbol{X}^\top \boldsymbol{X}\boldsymbol{\beta})(\xi - \boldsymbol{x}^\top \boldsymbol{W} \boldsymbol{X}^\top \boldsymbol{\xi}) + (\xi - \boldsymbol{x}^\top \boldsymbol{W} \boldsymbol{X}^\top \boldsymbol{\xi})^2\right] \\
&= \mathbb{E}\left[(\boldsymbol{x}^\top \boldsymbol{\beta} - \boldsymbol{x}^\top \boldsymbol{W} \boldsymbol{X}^\top \boldsymbol{X}\boldsymbol{\beta})^2 + (\xi - \boldsymbol{x}^\top \boldsymbol{W} \boldsymbol{X}^\top \boldsymbol{\xi})^2\right] + 2\mathbb{E}[(\boldsymbol{x}^\top \boldsymbol{\beta} - \boldsymbol{x}^\top \boldsymbol{W} \boldsymbol{X}^\top \boldsymbol{X}\boldsymbol{\beta})(\xi - \boldsymbol{x}^\top \boldsymbol{W} \boldsymbol{X}^\top \boldsymbol{\xi})] \\
&= \mathbb{E}\left[(\boldsymbol{x}^\top \boldsymbol{\beta} - \boldsymbol{x}^\top \boldsymbol{W} \boldsymbol{X}^\top \boldsymbol{X}\boldsymbol{\beta})^2 + (\xi - \boldsymbol{x}^\top \boldsymbol{W} \boldsymbol{X}^\top \boldsymbol{\xi})^2\right] \\
&= \underbrace{\mathbb{E}\left[(\boldsymbol{x}^\top \boldsymbol{W} \boldsymbol{X}^\top \boldsymbol{X}\boldsymbol{\beta})^2 + (\boldsymbol{x}^\top \boldsymbol{W} \boldsymbol{X}^\top \boldsymbol{\xi})^2\right]}_{f_1(\boldsymbol{W})} \underbrace{- 2\mathbb{E}[\boldsymbol{\beta}^\top \boldsymbol{x}\boldsymbol{x}^\top \boldsymbol{W} \boldsymbol{X}^\top \boldsymbol{X}\boldsymbol{\beta} + \xi \boldsymbol{x}^\top \boldsymbol{W} \boldsymbol{X}^\top \boldsymbol{\xi}]}_{f_2(\boldsymbol{W})} + \underbrace{\mathbb{E}[(\boldsymbol{x}^\top \boldsymbol{\beta})^2 + \xi^2]}_{\text{constant}}
\end{aligned}
\tag{25}
$$

where (25) follows Assumption 2.2. Since $f_2(\boldsymbol{W})$ is convex, $\mathcal{L}_{\text{PGD}}(\boldsymbol{W})$ is strongly-convex if and only if $f_1(\boldsymbol{W})$ is strongly-convex, which completes the proof of strong convexity.

Next, (20) and (21) in the proof of Lemma B.1 demonstrate that the optimal loss is achievable and is achieved at $\varepsilon = 0$. Subsequently, (17) indicates that $g_{\text{ATT}}^\star$ has the same form as $g_{\text{PGD}}^\star$. Under the strong convexity assumption, $g_{\text{PGD}}^\star$ is unique, which leads to the conclusion that $g_{\text{PGD}}^\star = g_{\text{ATT}}^\star$. □

### B.3. Proof of Lemma 2.5

*Proof.* According to Lemma 2.4, $\mathcal{L}_{\text{PGD}}(\mathcal{W})$ is strongly-convex as long as either $\mathbb{E}[(\boldsymbol{x}^\top \boldsymbol{W} \boldsymbol{X}^\top \boldsymbol{X} \boldsymbol{\beta})^2]$ or $\mathbb{E}[(\boldsymbol{x}^\top \boldsymbol{W} \boldsymbol{X}^\top \boldsymbol{\xi})^2]$ is strongly-convex. Therefore, in this lemma, the two claims correspond to the strong convexity of $\mathbb{E}[(\boldsymbol{x}^\top \boldsymbol{W} \boldsymbol{X}^\top \boldsymbol{\xi})^2]$ and $\mathbb{E}[(\boldsymbol{x}^\top \boldsymbol{W} \boldsymbol{X}^\top \boldsymbol{X} \boldsymbol{\beta})^2]$ terms, respectively.

Suppose the decomposition claim holds. Without losing generality, we may assume $(\boldsymbol{x}_1, \boldsymbol{\beta}_1, \boldsymbol{X}_1)$ are zero-mean because we can allocate the mean component to $(\boldsymbol{x}_2, \boldsymbol{\beta}_2, \boldsymbol{X}_2)$ without changing the covariance.

• **Claim 1:** Let $\bar{\boldsymbol{\Sigma}}_x = \mathbb{E}[\boldsymbol{x}_1 \boldsymbol{x}_1^\top]$, $\bar{\boldsymbol{\Sigma}}_\beta = \mathbb{E}[\boldsymbol{\beta}_1 \boldsymbol{\beta}_1^\top]$, and $\bar{\boldsymbol{\Sigma}}_X = \mathbb{E}[\boldsymbol{X}_1^\top \boldsymbol{X}_1]$. If the first claim holds, using independence, observe that we can write

$$\mathbb{E}[(\boldsymbol{x}^\top \boldsymbol{W} \boldsymbol{X}^\top \boldsymbol{\xi})^2] = \mathbb{E}[(\boldsymbol{x}_1^\top \boldsymbol{W} \boldsymbol{X}_1^\top \boldsymbol{\xi})^2] + \mathbb{E}[(\boldsymbol{x}_1^\top \boldsymbol{W} \boldsymbol{X}_2^\top \boldsymbol{\xi})^2] + \mathbb{E}[(\boldsymbol{x}_2^\top \boldsymbol{W} \boldsymbol{X}_1^\top \boldsymbol{\xi})^2] + \mathbb{E}[(\boldsymbol{x}_2^\top \boldsymbol{W} \boldsymbol{X}_2^\top \boldsymbol{\xi})^2],$$

where the last three terms of the right hand side are convex and the first term obeys

$$\begin{aligned}
\mathbb{E}[(\boldsymbol{x}_1^\top \boldsymbol{W} \boldsymbol{X}_1^\top \boldsymbol{\xi})^2] &= \sigma^2 \, \mathbb{E}[\boldsymbol{x}_1^\top \boldsymbol{W} \boldsymbol{X}_1^\top \boldsymbol{X}_1 \boldsymbol{W}^\top \boldsymbol{x}_1] \\
&= \sigma^2 \text{tr}\left(\mathbb{E}[\boldsymbol{x}_1 \boldsymbol{x}_1^\top \boldsymbol{W} \boldsymbol{X}_1^\top \boldsymbol{X}_1 \boldsymbol{W}^\top]\right) \\
&= \sigma^2 \text{tr}\left(\bar{\boldsymbol{\Sigma}}_x \boldsymbol{W} \bar{\boldsymbol{\Sigma}}_X \boldsymbol{W}^\top\right) \\
&= \sigma^2 \left\| \sqrt{\bar{\boldsymbol{\Sigma}}_x} \boldsymbol{W} \sqrt{\bar{\boldsymbol{\Sigma}}_X} \right\|_F^2.
\end{aligned}$$

Since noise level $\sigma > 0$, using the full-rankness of covariance matrices $\bar{\boldsymbol{\Sigma}}_x$ and $\bar{\boldsymbol{\Sigma}}_X$, we conclude with strong convexity of $\mathbb{E}[(\boldsymbol{x}^\top \boldsymbol{W} \boldsymbol{X}^\top \boldsymbol{\xi})^2]$.

• **Claim 2:** Now recall that $\bar{\boldsymbol{\Sigma}}_X = \mathbb{E}[\boldsymbol{X}_1^\top \boldsymbol{X}_1]$ and set $\boldsymbol{A} = \boldsymbol{X}_1^\top \boldsymbol{X}_1 - \bar{\boldsymbol{\Sigma}}_X$ and $\boldsymbol{B} = \boldsymbol{X}_2^\top \boldsymbol{X}_2 + \bar{\boldsymbol{\Sigma}}_X$. Observe that $\mathbb{E}[\boldsymbol{A}] = 0$. If the second claim holds, $\mathbb{E}[\boldsymbol{X}^\top \boldsymbol{X}] = \mathbb{E}[\boldsymbol{A} + \boldsymbol{B}]$. Note that $(\boldsymbol{A}, \boldsymbol{\beta}_1, \boldsymbol{x}_1)$ are independent of each other and $(\boldsymbol{B}, \boldsymbol{\beta}_2, \boldsymbol{x}_2)$. Using independence and $\mathbb{E}[\boldsymbol{A}] = 0$, similarly write

$$\mathbb{E}[(\boldsymbol{x}^\top \boldsymbol{W} \boldsymbol{X}^\top \boldsymbol{X} \boldsymbol{\beta})^2] = \mathbb{E}[(\boldsymbol{x}^\top \boldsymbol{W} \boldsymbol{A} \boldsymbol{\beta})^2] + \mathbb{E}[(\boldsymbol{x}^\top \boldsymbol{W} \boldsymbol{B} \boldsymbol{\beta})^2].$$

Now using $\mathbb{E}[\boldsymbol{\beta}_1] = \mathbb{E}[\boldsymbol{x}_1] = 0$ and their independence from rest, these terms obeys

$$\begin{aligned}
\mathbb{E}[(\boldsymbol{x}^\top \boldsymbol{W} \boldsymbol{A} \boldsymbol{\beta})^2] &= \mathbb{E}[(\boldsymbol{x}_1^\top \boldsymbol{W} \boldsymbol{A} \boldsymbol{\beta}_1)^2] + \mathbb{E}[(\boldsymbol{x}_1^\top \boldsymbol{W} \boldsymbol{A} \boldsymbol{\beta}_2)^2] + \mathbb{E}[(\boldsymbol{x}_2^\top \boldsymbol{W} \boldsymbol{A} \boldsymbol{\beta}_1)^2] + \mathbb{E}[(\boldsymbol{x}_2^\top \boldsymbol{W} \boldsymbol{A} \boldsymbol{\beta}_2)^2] \\
\mathbb{E}[(\boldsymbol{x}^\top \boldsymbol{W} \boldsymbol{B} \boldsymbol{\beta})^2] &= \mathbb{E}[(\boldsymbol{x}_1^\top \boldsymbol{W} \boldsymbol{B} \boldsymbol{\beta}_1)^2] + \mathbb{E}[(\boldsymbol{x}_1^\top \boldsymbol{W} \boldsymbol{B} \boldsymbol{\beta}_2)^2] + \mathbb{E}[(\boldsymbol{x}_2^\top \boldsymbol{W} \boldsymbol{B} \boldsymbol{\beta}_1)^2] + \mathbb{E}[(\boldsymbol{x}_2^\top \boldsymbol{W} \boldsymbol{B} \boldsymbol{\beta}_2)^2].
\end{aligned}$$

In both equations, the last three terms of the right hand side are convex. To proceed, we focus on the first terms. Using independence and setting $\boldsymbol{\Sigma}_X = \mathbb{E}[\boldsymbol{X}^\top \boldsymbol{X}] \geq \bar{\boldsymbol{\Sigma}}_X > 0$, we note that

$$\mathbb{E}[(\boldsymbol{x}_1^\top \boldsymbol{W} \boldsymbol{A} \boldsymbol{\beta}_1)^2] + \mathbb{E}[(\boldsymbol{x}_1^\top \boldsymbol{W} \boldsymbol{B} \boldsymbol{\beta}_1)^2] = \mathbb{E}[(\boldsymbol{x}_1^\top \boldsymbol{W} \boldsymbol{X}^\top \boldsymbol{X} \boldsymbol{\beta}_1)^2]$$

where $\boldsymbol{x}_1, \boldsymbol{\beta}_1, \boldsymbol{X}$ are independent and full-rank covariance. To proceed, note that

$$\mathbb{E}[(\boldsymbol{x}_1^\top \boldsymbol{W} \boldsymbol{X}^\top \boldsymbol{X} \boldsymbol{\beta}_1)^2] = \mathbb{E}[(\boldsymbol{x}_1^\top \boldsymbol{W} \boldsymbol{\Sigma}_X \boldsymbol{\beta}_1)^2] + \mathbb{E}[(\boldsymbol{x}_1^\top \boldsymbol{W} (\boldsymbol{X}^\top \boldsymbol{X} - \boldsymbol{\Sigma}_X) \boldsymbol{\beta}_1)^2].$$

Observing the convexity of the right hand side and focusing on the first term, we get

$$\mathbb{E}[(\boldsymbol{x}_1^\top \boldsymbol{W} \boldsymbol{\Sigma}_X \boldsymbol{\beta}_1)^2] = \text{tr}\left(\bar{\boldsymbol{\Sigma}}_x \boldsymbol{W} \boldsymbol{\Sigma}_X \bar{\boldsymbol{\Sigma}}_\beta \boldsymbol{\Sigma}_X \boldsymbol{W}^\top\right) = \left\| \sqrt{\bar{\boldsymbol{\Sigma}}_x} \boldsymbol{W} \boldsymbol{\Sigma}_X \sqrt{\bar{\boldsymbol{\Sigma}}_\beta} \right\|_F^2.$$

Using the fact that covariance matrices, $\bar{\boldsymbol{\Sigma}}_x, \boldsymbol{\Sigma}_X, \bar{\boldsymbol{\Sigma}}_\beta$, are full rank concludes the strong convexity proof of $\mathbb{E}[(\boldsymbol{x}^\top \boldsymbol{W} \boldsymbol{X}^\top \boldsymbol{X} \boldsymbol{\beta})^2]$.

□

## C. Analysis of General Data Distribution

In this section, we provide the proofs in Section 3, which focuses on solving Objective (5a). For the sake of clean notation, let $\mathcal{L}(\boldsymbol{W}) := \mathcal{L}_{\text{PGD}}(\boldsymbol{W})$ and $g := g_{\text{PGD}}$ in this section.

## C.1. Supporting Results

We begin by deriving the even moments of random variables.

- **$2n$'th moment of a normally distributed variable:** Let $u \sim \mathcal{N}(0, \sigma^2)$. Then we have

$$\mathbb{E}[u^{2n}] = \sigma^{2n}(2n-1)!!. \tag{26}$$

- **4'th moment:** Let $\boldsymbol{u} \sim \mathcal{N}(0, \boldsymbol{I}_d)$. Then for any $\boldsymbol{W}, \boldsymbol{W}' \in \mathbb{R}^{d \times d}$, we have

$$
\begin{aligned}
&\mathbb{E}\left[(\boldsymbol{u}^\top \boldsymbol{W} \boldsymbol{u})(\boldsymbol{u}^\top \boldsymbol{W}' \boldsymbol{u})\right] \\
&= \mathbb{E}\left[\left(\sum_{i,j=1}^d W_{ij} u_i u_j\right)\left(\sum_{i,j=1}^d W'_{ij} u_i u_j\right)\right] \\
&= \mathbb{E}\left[\left(\sum_{i=1}^d W_{ii} u_i^2\right)\left(\sum_{i=1}^d W'_{ii} u_i^2\right)\right] + \mathbb{E}\left[\left(\sum_{i \neq j} W_{ij} u_i u_j\right)\left(\sum_{i \neq j} W'_{ij} u_i u_j\right)\right] \\
&= \sum_{i=1}^d W_{ii} W'_{ii} \mathbb{E}\left[u_i^4\right] + \sum_{i \neq j} W_{ii} W'_{jj} \mathbb{E}[u_i^2]\mathbb{E}[u_j^2] + \sum_{i \neq j} W_{ij} W'_{ij} \mathbb{E}[u_i^2]\mathbb{E}[u_j^2] + \sum_{i \neq j} W_{ij} W'_{ji} \mathbb{E}[u_i^2]\mathbb{E}[u_j^2] \\
&= 3\sum_{i=1}^d W_{ii} W'_{ii} + \sum_{i \neq j} W_{ii} W'_{jj} + \sum_{i \neq j} W_{ij} W'_{ij} + \sum_{i \neq j} W_{ij} W'_{ji} \\
&= \sum_{i,j=1}^d W_{ii} W'_{jj} + \sum_{i,j=1}^d W_{ij} W'_{ij} + \sum_{i,j=1}^d W_{ij} W'_{ji} \\
&= \operatorname{tr}(\boldsymbol{W}) \operatorname{tr}(\boldsymbol{W}') + \operatorname{tr}\left(\boldsymbol{W}' \boldsymbol{W}^\top\right) + \operatorname{tr}(\boldsymbol{W} \boldsymbol{W}').
\end{aligned} \tag{27}
$$

- **4'th cross-moment:** Let $\boldsymbol{u}, \boldsymbol{v} \sim \mathcal{N}(0, \boldsymbol{I}_d)$ and for any $\boldsymbol{W} \in \mathbb{R}^{d \times d}$, let $\boldsymbol{\Lambda_W} = \boldsymbol{W} \odot \boldsymbol{I}_d$. Then we have

$$
\begin{aligned}
&\mathbb{E}\left[(\boldsymbol{u}^\top \boldsymbol{W} \boldsymbol{v} \boldsymbol{v}^\top \boldsymbol{u})^2\right] \\
&= \mathbb{E}\left[\left(\sum_{i,j=1}^d W_{ij} u_i v_j\right)^2 \left(\sum_{i=1}^d u_i v_i\right)^2\right] \\
&= \mathbb{E}\left[\left(\sum_{i,j=1}^d W_{ij}^2 u_i^2 v_j^2 + \sum_{i \neq i'} W_{ij} W_{i'j} u_i u_{i'} v_j^2 + \sum_{j \neq j'} W_{ij} W_{ij'} u_i^2 v_j v_{j'} + \sum_{i' \neq i, j' \neq j} W_{ij} W_{i'j'} u_i u_{i'} v_j v_{j'}\right)\left(\sum_{i=1}^d u_i^2 v_i^2 + \sum_{i \neq j} u_i u_j v_i v_j\right)\right] \\
&= \mathbb{E}\left[\left(\sum_{i,j=1}^d W_{ij}^2 u_i^2 v_j^2\right)\left(\sum_{i=1}^d u_i^2 v_i^2\right) + \left(\sum_{i \neq j} W_{ij} W_{ji} u_i^2 u_j^2 v_i^2 v_j^2\right)\right] \\
&= \mathbb{E}\left[\left(\sum_{i=1}^d W_{ii}^2 u_i^2 v_i^2 + \sum_{i \neq j} W_{ij}^2 u_i^2 v_j^2\right)\left(\sum_{i=1}^d u_i^2 v_i^2\right)\right] + \sum_{i \neq j} W_{ij} W_{ji} \\
&= \mathbb{E}\left[\left(\sum_{i=1}^d W_{ii}^2 u_i^4 v_i^4 + \sum_{i \neq j} W_{ii}^2 u_i^2 v_i^2 u_j^2 v_j^2\right)\right] + \mathbb{E}\left[\left(\sum_{i \neq j} W_{ij}^2 u_i^4 v_j^2 v_i^2 + \sum_{i \neq j} W_{ij}^2 u_i^2 v_j^4 u_j^2 + \sum_{i \neq j \neq k} W_{ij}^2 u_i^2 v_j^2 u_k^2 v_k^2\right)\right] + \sum_{i \neq j} W_{ij} W_{ji} \\
&= 9\sum_{i=1}^d W_{ii}^2 + (d-1)\sum_{i=1}^d W_{ii}^2 + 6\sum_{i \neq j} W_{ij}^2 + (d-2)\sum_{i \neq j} W_{ij}^2 + \sum_{i \neq j} W_{ij} W_{ji} \\
&= 3\sum_{i=1}^d W_{ii}^2 + (d+4)\sum_{i,j=1}^d W_{ij}^2 + \sum_{i,j=1}^d W_{ij} W_{ji} \\
&= 3\operatorname{tr}\left(\boldsymbol{\Lambda_W}^2\right) + (d+4)\operatorname{tr}\left(\boldsymbol{W}\boldsymbol{W}^\top\right) + \operatorname{tr}\left(\boldsymbol{W}^2\right).
\end{aligned} \tag{28}
$$

- **6'th moment:** Let $\boldsymbol{u} \sim \mathcal{N}(0, \boldsymbol{I}_d)$. Then for any $\boldsymbol{W}, \boldsymbol{W}' \in \mathbb{R}^{d \times d}$, we have

$$
\mathbb{E}\left[(\boldsymbol{u}^\top \boldsymbol{W} \boldsymbol{u})(\boldsymbol{u}^\top \boldsymbol{W}' \boldsymbol{u}) \|\boldsymbol{u}\|_{\ell_2}^2\right]
$$

$$
= \mathbb{E}\left[\left(\sum_{i,j=1}^d W_{ij} u_i u_j\right)\left(\sum_{i,j=1}^d W_{ij}' u_i u_j\right)\left(\sum_{i=1}^d u_i^2\right)\right]
$$

$$
= \mathbb{E}\left[\left(\sum_{i=1}^d W_{ii} u_i^2\right)\left(\sum_{i=1}^d W_{ii}' u_i^2\right)\left(\sum_{i=1}^d u_i^2\right)\right] + \mathbb{E}\left[\left(\sum_{i \neq j} W_{ij} u_i u_j\right)\left(\sum_{i \neq j} W_{ij}' u_i u_j\right)\left(\sum_{i=1}^d u_i^2\right)\right]
$$

$$
= \sum_{i=1}^d W_{ii} W_{ii}' \mathbb{E}\left[u_i^4\left(\sum_{i'=1}^d u_{i'}^2\right)\right] + \sum_{i \neq j} W_{ii} W_{jj}' \mathbb{E}\left[u_i^2 u_j^2\left(\sum_{i'=1}^d u_{i'}^2\right)\right]
$$

$$
+ \sum_{i \neq j} W_{ij} W_{ij}' \mathbb{E}\left[u_i^2 u_j^2\left(\sum_{i'=1}^d u_{i'}^2\right)\right] + \sum_{i \neq j} W_{ij} W_{ji}' \mathbb{E}\left[u_i^2 u_j^2\left(\sum_{i'=1}^d u_{i'}^2\right)\right]
$$

$$
= (d + 4)\left(3 \sum_{i=1}^d W_{ii} W_{ii}' + \sum_{i \neq j} W_{ii} W_{jj}' + \sum_{i \neq j} W_{ij} W_{ij}' + \sum_{i \neq j} W_{ij} W_{ji}'\right) \tag{29}
$$

$$
= (d + 4)\left(\sum_{i,j=1}^d W_{ii} W_{jj}' + \sum_{i,j=1}^d W_{ij} W_{ij}' + \sum_{i,j=1}^d W_{ij} W_{ji}'\right)
$$

$$
= (d + 4)\left(\operatorname{tr}(\boldsymbol{W}) \operatorname{tr}(\boldsymbol{W}') + \operatorname{tr}\left(\boldsymbol{W}' \boldsymbol{W}^\top\right) + \operatorname{tr}(\boldsymbol{W} \boldsymbol{W}')\right), \tag{30}
$$

where (29) is obtained by following

$$
\mathbb{E}\left[u_i^4\left(\sum_{i'=1}^d u_{i'}^2\right)\right] = \mathbb{E}[u^6] + (d - 1)\mathbb{E}[u^4]\mathbb{E}[u^2] = 3(d + 4),
$$

$$
\mathbb{E}\left[u_i^2 u_j^2\left(\sum_{i'=1}^d u_{i'}^2\right)\right] = 2\mathbb{E}[u^4]\mathbb{E}[u^2] + (d - 2)\mathbb{E}[u^2]\mathbb{E}[u^2]\mathbb{E}[u^2] = d + 4.
$$

- **8'th moment:** Let $\boldsymbol{u} \sim \mathcal{N}(0, \boldsymbol{I}_d)$. Then for any $\boldsymbol{W}, \boldsymbol{W}' \in \mathbb{R}^{d \times d}$, we have

$$
\mathbb{E}\left[(\boldsymbol{u}^\top \boldsymbol{W} \boldsymbol{u})(\boldsymbol{u}^\top \boldsymbol{W}' \boldsymbol{u}) \|\boldsymbol{u}\|_{\ell_2}^4\right]
$$

$$
= \mathbb{E}\left[\left(\sum_{i,j=1}^d W_{ij} u_i u_j\right)\left(\sum_{i,j=1}^d W_{ij}' u_i u_j\right)\left(\sum_{i,j=1}^d u_i^2 u_j^2\right)\right]
$$

$$
= \mathbb{E}\left[\left(\sum_{i=1}^d W_{ii} u_i^2\right)\left(\sum_{i=1}^d W_{ii}' u_i^2\right)\left(\sum_{i=1}^d u_i^4 + \sum_{i \neq j} u_i^2 u_j^2\right)\right] + \mathbb{E}\left[\left(\sum_{i \neq j} W_{ij} u_i u_j\right)\left(\sum_{i \neq j} W_{ij}' u_i u_j\right)\left(\sum_{i=1}^d u_i^4 + \sum_{i \neq j} u_i^2 u_j^2\right)\right]
$$

$$
= \sum_{i=1}^d W_{ii} W_{ii}' \mathbb{E}\left[u_i^4\left(\sum_{i'=1}^d u_{i'}^4 + \sum_{i' \neq j'} u_{i'}^2 u_{j'}^2\right)\right] + \sum_{i \neq j} W_{ii} W_{jj}' \mathbb{E}\left[u_i^2 u_j^2\left(\sum_{i'=1}^d u_{i'}^4 + \sum_{i' \neq j'} u_{i'}^2 u_{j'}^2\right)\right]
$$

$$
+ \sum_{i \neq j} W_{ij} W_{ij}' \mathbb{E}\left[u_i^2 u_j^2\left(\sum_{i'=1}^d u_{i'}^4 + \sum_{i' \neq j'} u_{i'}^2 u_{j'}^2\right)\right] + \sum_{i \neq j} W_{ij} W_{ji}' \mathbb{E}\left[u_i^2 u_j^2\left(\sum_{i'=1}^d u_{i'}^4 + \sum_{i' \neq j'} u_{i'}^2 u_{j'}^2\right)\right]
$$

$$
= (d + 4)(d + 6)\left(3 \sum_{i=1}^d W_{ii} W_{ii}' + \sum_{i \neq j} W_{ii} W_{jj}' + \sum_{i \neq j} W_{ij} W_{ij}' + \sum_{i \neq j} W_{ij} W_{ji}'\right) \tag{31}
$$

$$
= (d + 4)(d + 6)\left(\sum_{i,j=1}^d W_{ii} W_{jj}' + \sum_{i,j=1}^d W_{ij} W_{ij}' + \sum_{i,j=1}^d W_{ij} W_{ji}'\right)
$$

$$
= (d + 4)(d + 6)\left(\operatorname{tr}(\boldsymbol{W}) \operatorname{tr}(\boldsymbol{W}') + \operatorname{tr}\left(\boldsymbol{W}' \boldsymbol{W}^\top\right) + \operatorname{tr}(\boldsymbol{W} \boldsymbol{W}')\right). \tag{32}
$$

where (31) is obtained by following

$$\mathbb{E}\left[u_i^4\left(\sum_{i'=1}^d u_{i'}^4 + \sum_{i'\neq j'} u_{i'}^2 u_{j'}^2\right)\right]$$

$$= \mathbb{E}[u^8] + (d-1)\mathbb{E}[u^4]\mathbb{E}[u^4] + 2(d-1)\mathbb{E}[u^6]\mathbb{E}[u^2] + (d-1)(d-2)\mathbb{E}[u^4]\mathbb{E}[u^2]\mathbb{E}[u^2]$$

$$= 105 + 9(d-1) + 30(d-1) + 3(d-1)(d-2)$$

$$= 3(d+4)(d+6),$$

$$\mathbb{E}\left[u_i^2 u_j^2\left(\sum_{i'=1}^d u_{i'}^4 + \sum_{i'\neq j'} u_{i'}^2 u_{j'}^2\right)\right]$$

$$= 2\mathbb{E}[u^6]\mathbb{E}[u^2] + (d-2)\mathbb{E}[u^4](\mathbb{E}[u^2])^2 + 2\mathbb{E}[u^4]\mathbb{E}[u^4] + 4(d-2)\mathbb{E}[u^4](\mathbb{E}[u^2])^2 + (d-2)(d-3)(\mathbb{E}[u^2])^4$$

$$= 30 + 3(d-2) + 18 + 12(d-2) + (d-2)(d-3)$$

$$= (d+4)(d+6).$$

## C.2. Independent Data with General Covariance

*Proof of Theorem 3.1.* Consider a general independent linear model as defined in (7) where $\Sigma_x$ and $\Sigma_\beta$ are full-rank feature and task convariance matrices and

$$\boldsymbol{x} \sim \mathcal{N}(0, \Sigma_x), \quad \boldsymbol{\beta} \sim \mathcal{N}(0, \Sigma_\beta), \quad \xi \sim \mathcal{N}(0, \sigma^2), \quad \text{and} \quad y = \boldsymbol{x}^\top \boldsymbol{\beta} + \xi.$$

Let

$$X = [\boldsymbol{x}_1 \cdots \boldsymbol{x}_n]^\top, \quad \boldsymbol{\xi} = [\xi_1 \cdots \xi_n]^\top, \quad \text{and} \quad \boldsymbol{y} = [y_1 \cdots y_n]^\top = X\boldsymbol{\beta} + \boldsymbol{\xi}.$$

To simplify and without loss of generality, let $\bar{\boldsymbol{x}} = \Sigma_x^{-1/2}\boldsymbol{x}$, $\bar{X} = X\Sigma_x^{-1/2}$, $\bar{\boldsymbol{\beta}} = \Sigma_x^{1/2}\boldsymbol{\beta}$ where we have

$$\bar{\boldsymbol{x}} \sim \mathcal{N}(0, \boldsymbol{I}), \qquad \bar{\boldsymbol{\beta}} \sim \mathcal{N}(0, \Sigma_x^{1/2}\Sigma_\beta\Sigma_x^{1/2})$$

and

$$y = \bar{\boldsymbol{x}}^\top \bar{\boldsymbol{\beta}} + \xi, \qquad \boldsymbol{y} = \bar{X}\bar{\boldsymbol{\beta}} + \boldsymbol{\xi}.$$

Then recap the loss from (5a), and we obtain

$$\mathcal{L}(\boldsymbol{W}) = \mathbb{E}\left[(y - g(\boldsymbol{Z}))^2\right]$$

$$= \mathbb{E}\left[\left(\boldsymbol{x}^\top\boldsymbol{\beta} + \xi - \boldsymbol{x}^\top \boldsymbol{W}X^\top(X\boldsymbol{\beta} + \boldsymbol{\xi})\right)^2\right]$$

$$= \mathbb{E}\left[(\boldsymbol{x}^\top\boldsymbol{\beta} - \boldsymbol{x}^\top \boldsymbol{W}X^\top X\boldsymbol{\beta})^2 + 2(\boldsymbol{x}^\top\boldsymbol{\beta} - \boldsymbol{x}^\top \boldsymbol{W}X^\top X\boldsymbol{\beta})(\xi - \boldsymbol{x}^\top \boldsymbol{W}X^\top\boldsymbol{\xi}) + (\xi - \boldsymbol{x}^\top \boldsymbol{W}X^\top\boldsymbol{\xi})^2\right]$$

$$= \mathbb{E}\left[(\boldsymbol{x}^\top\boldsymbol{\beta} - \boldsymbol{x}^\top \boldsymbol{W}X^\top X\boldsymbol{\beta})^2\right] + \mathbb{E}\left[(\boldsymbol{x}^\top \boldsymbol{W}X^\top\boldsymbol{\xi})^2\right] + \sigma^2, \tag{33}$$

where the last equality comes from the independence of label noise $\xi, \boldsymbol{\xi}$.

We first consider the following term

$$\mathbb{E}\left[(\boldsymbol{x}^\top \boldsymbol{W}X^\top\boldsymbol{\xi})^2\right] = \mathbb{E}\left[(\bar{\boldsymbol{x}}^\top(\Sigma_x^{1/2}\boldsymbol{W}\Sigma_x^{1/2})\bar{X}^\top\boldsymbol{\xi})^2\right] = n\sigma^2 \cdot \text{tr}\left(\bar{\boldsymbol{W}}\bar{\boldsymbol{W}}^\top\right)$$

where we define $\bar{\boldsymbol{W}} = \Sigma_x^{1/2}\boldsymbol{W}\Sigma_x^{1/2}$. Next, focus on the following

$$\mathbb{E}\left[(\boldsymbol{x}^\top\boldsymbol{\beta} - \boldsymbol{x}^\top \boldsymbol{W}X^\top X\boldsymbol{\beta})^2\right] = \mathbb{E}\left[(\bar{\boldsymbol{x}}^\top\bar{\boldsymbol{\beta}} - \bar{\boldsymbol{x}}^\top\bar{\boldsymbol{W}}\bar{X}^\top\bar{X}\bar{\boldsymbol{\beta}})^2\right]$$

$$= \mathbb{E}\left[\left(\bar{\boldsymbol{x}}^\top\left(\boldsymbol{I} - \bar{\boldsymbol{W}}\bar{X}^\top\bar{X}\right)\bar{\boldsymbol{\beta}}\right)^2\right]$$

$$= \text{tr}\left(\mathbb{E}\left[\left(\boldsymbol{I} - \bar{\boldsymbol{W}}\bar{X}^\top\bar{X}\right)\Sigma\left(\boldsymbol{I} - \bar{\boldsymbol{W}}\bar{X}^\top\bar{X}\right)^\top\right]\right)$$

$$= \text{tr}\left(\Sigma\right) - \text{tr}\left(\Sigma(\bar{\boldsymbol{W}} + \bar{\boldsymbol{W}}^\top)\mathbb{E}[\bar{X}^\top\bar{X}]\right) + \text{tr}\left(\bar{\boldsymbol{W}}^\top\bar{\boldsymbol{W}}\mathbb{E}[\bar{X}^\top\bar{X}\Sigma\bar{X}^\top\bar{X}]\right)$$

$$= \text{tr}\left(\Sigma\right) - 2n\cdot\text{tr}\left(\Sigma\bar{\boldsymbol{W}}\right) + \text{tr}\left(\bar{\boldsymbol{W}}^\top\bar{\boldsymbol{W}}\mathbb{E}[\bar{X}^\top\bar{X}\Sigma\bar{X}^\top\bar{X}]\right),$$

where $\Sigma := \Sigma_x^{1/2} \Sigma_\beta \Sigma_x^{1/2}$.

Let $\bar{x}_i \in \mathbb{R}^n$ be the $i$'th column of $\bar{X}$ and $\Sigma_{ij}$ be the $(i, j)$'th entry of $\Sigma$. Then the $(i, j)$ entry of matrix $\bar{X}^\top \bar{X} \Sigma \bar{X}^\top \bar{X}$ is

$$(\bar{X}^\top \bar{X} \Sigma \bar{X}^\top \bar{X})_{ij} = \sum_{k=1}^d \sum_{p=1}^d \Sigma_{kp} \bar{x}_i^\top \bar{x}_k \bar{x}_p^\top \bar{x}_j.$$

Then we get

$$i \neq j: \quad \mathbb{E}\left[\left(\bar{X}^\top \bar{X} \Sigma \bar{X}^\top \bar{X}\right)_{ij}\right] = \Sigma_{ij} \mathbb{E}[\bar{x}_i^\top \bar{x}_i \bar{x}_j^\top \bar{x}_j] + \Sigma_{ji} \mathbb{E}[\bar{x}_i^\top \bar{x}_j \bar{x}_i^\top \bar{x}_j] = n^2 \Sigma_{ij} + n\Sigma_{ji}$$

$$i = j: \quad \mathbb{E}\left[\left(\bar{X}^\top \bar{X} \Sigma \bar{X}^\top \bar{X}\right)_{ii}\right] = \Sigma_{ii} \mathbb{E}\left[\bar{x}_i^\top \bar{x}_i \bar{x}_i^\top \bar{x}_i\right] + \sum_{j \neq i} \Sigma_{jj} \mathbb{E}\left[\bar{x}_i^\top \bar{x}_j \bar{x}_j^\top \bar{x}_i\right]$$

$$= \Sigma_{ii} \mathbb{E}\left[(x_{i1}^2 + \cdots + x_{in}^2)^2\right] + n \sum_{j \neq i} \Sigma_{jj}$$

$$= \Sigma_{ii}(3n + n(n - 1)) + n \sum_{j \neq i} \Sigma_{jj}$$

$$= n\left(\Sigma_{ii}(n + 1) + \sum_{j=1}^d \Sigma_{jj}\right)$$

$$= n\left(\Sigma_{ii}(n + 1) + \mathrm{tr}\,(\Sigma)\right).$$

Therefore

$$\mathbb{E}[\bar{X}^\top \bar{X} \Sigma \bar{X}^\top \bar{X}] = n(n + 1)\Sigma + n \cdot \mathrm{tr}\,(\Sigma)\, I.$$

Combining all together results in

$$\mathcal{L}(W) = \mathrm{tr}\,(\Sigma) - 2n\mathrm{tr}\left(\Sigma \bar{W}\right) + n(n + 1)\mathrm{tr}\left(\Sigma \bar{W}^\top \bar{W}\right) + n(\mathrm{tr}\,(\Sigma) + \sigma^2)\mathrm{tr}\left(\bar{W} \bar{W}^\top\right) + \sigma^2,$$
$$= M - 2n\mathrm{tr}\left(\Sigma \bar{W}\right) + n(n + 1)\mathrm{tr}\left(\Sigma \bar{W}^\top \bar{W}\right) + nM\mathrm{tr}\left(\bar{W} \bar{W}^\top\right), \tag{34}$$

where $M := \mathrm{tr}\,(\Sigma) + \sigma^2$. Setting $\nabla_{\bar{W}} \mathcal{L}(W) = 0$ returns

$$-2n \cdot \Sigma + 2n(n + 1) \cdot \Sigma \bar{W} + 2nM\bar{W} = 0 \implies \bar{W}_\star = \left((n + 1)I + M\Sigma^{-1}\right)^{-1}.$$

Then we have

$$W_\star = \Sigma_x^{-1/2}\left((n + 1)I + M\Sigma^{-1}\right)^{-1} \Sigma_x^{-1/2}$$

and

$$\mathcal{L}_\star = \mathcal{L}(W_\star) = M - n\mathrm{tr}\left(((n + 1)\Sigma^{-1} + M\Sigma^{-2})^{-1}\right).$$

$\square$

## C.3. Retrieval Augmented Generation with $\alpha$ Correlation

In this section, we consider the retrieval augmented generation (RAG) linear model similar to (9), where we first draw the query vector $x$ and task vector $\beta$ via

$$x \sim \mathcal{N}(0, I) \quad \text{and} \quad \beta \sim \mathcal{N}(0, I).$$

We then draw data $(x_i)_{i=1}^n$ to be used in-context according to the rule corr_coef$(x, x_i) \geq \alpha \geq 0$. Hence, for $i \leq n$ we sample

$$x_i \mid x \sim \mathcal{N}(\alpha x, \gamma^2 I), \quad \xi_i \sim \mathcal{N}(0, \sigma^2) \quad \text{and} \quad y_i = x_i^\top \beta + \xi_i, \tag{35}$$

which results in (9) by setting $\gamma^2 = 1 - \alpha^2$.

**Theorem C.1** (Extended version of Theorem 3.3). *Consider linear model as defined in* (35). *Recap the objective from* (5a) *and let* $W_\star := \arg\min_W \mathcal{L}_{PGD}(W)$, *and* $\mathcal{L}_\star = \mathcal{L}_{PGD}(W_\star)$. *Then* $W_\star$ *and* $\mathcal{L}_\star$ *satisfy*

$$W_\star = cI \qquad and \qquad \mathcal{L}_\star = d + \sigma^2 - cnd(\alpha^2(d+2) + \gamma^2) \tag{36}$$

*where*

$$c = \frac{\alpha^2(d+2) + \gamma^2}{\alpha^4 n(d+2)(d+4) + \alpha^2\gamma^2(d+2)(d+2n+3) + \gamma^4(d+n+1) + \sigma^2(\alpha^2(d+2) + \gamma^2)}.$$

*Suppose* $\alpha = O\left(1/\sqrt{d}\right)$, $d/n = O(1)$ *and* $d$ *is sufficiently large. Let* $\kappa = \alpha^2 d + 1$ *and* $\gamma^2 = 1 - \alpha^2$. *Then* $W_\star$ *and* $\mathcal{L}_\star$ *have approximate forms*

$$W_\star \approx \frac{1}{\kappa n + d + \sigma^2} I \qquad and \qquad \mathcal{L}_\star \approx d + \sigma^2 - \frac{\kappa nd}{\kappa n + d + \sigma^2}. \tag{37}$$

*Proof.* Here, for clean notation and without loss of generality, we define and rewrite (35) via

$$g_i \sim \mathcal{N}(0, I), \quad \xi_i \sim \mathcal{N}(0, \sigma^2) \quad \text{and} \quad x_i = \alpha x + \gamma g_i, \quad y_i = (\alpha x + \gamma g_i)^\top \beta + \xi_i.$$

Then we obtain

$$\begin{aligned}
\mathcal{L}(W) &= \mathbb{E}\left[(y - g(Z))^2\right] \\
&= \mathbb{E}\left[\left(x^\top\beta + \xi - x^\top W X^\top(X\beta + \xi)\right)^2\right] \\
&= \mathbb{E}\left[(x^\top\beta - x^\top W X^\top X\beta)^2 + 2(x^\top\beta - x^\top W X^\top X\beta)(\xi - x^\top W X^\top \xi) + (\xi - x^\top W X^\top \xi)^2\right] \\
&= \mathbb{E}\left[(x^\top\beta - x^\top W X^\top X\beta)^2\right] + \mathbb{E}\left[(x^\top W X^\top \xi)^2\right] + \sigma^2.
\end{aligned} \tag{38}$$

To begin with, let

$$N_1 = \mathtt{tr}\,(W)^2 + \mathtt{tr}\left(WW^\top\right) + \mathtt{tr}\left(W^2\right), \quad N_2 = \mathtt{tr}\left(WW^\top\right), \quad \text{and} \quad N_3 = \mathtt{tr}\,(W).$$

We first focus on the second term in (38)

$$\begin{aligned}
\mathbb{E}\left[(x^\top W X^\top \xi)^2\right] &= \mathbb{E}\left[\left(\sum_{i=1}^n \xi_i x^\top W(\alpha x + \gamma g_i)\right)^2\right] \\
&= n\sigma^2 \,\mathbb{E}\left[x^\top W(\alpha x + \gamma g)(\alpha x + \gamma g)^\top W^\top x\right] \\
&= n\sigma^2 \left(\alpha^2 \,\mathbb{E}[x^\top W x x^\top W^\top x] + \gamma^2 \,\mathbb{E}[x^\top W g g^\top W^\top x]\right) \\
&= n\sigma^2 \left(\alpha^2 N_1 + \gamma^2 N_2\right). \qquad \text{(It follows (27) and independence of } x, g.)
\end{aligned}$$

Next, the first term in (38) can be decomposed into

$$\mathbb{E}\left[(x^\top\beta - x^\top W X^\top X\beta)^2\right] = \underbrace{\mathbb{E}\left[(x^\top\beta)^2\right]}_{(a)} + \underbrace{\mathbb{E}\left[(x^\top W X^\top X\beta)^2\right]}_{(b)} - \underbrace{2\mathbb{E}\left[x^\top\beta x^\top W X^\top X\beta\right]}_{(c)}.$$

In the following, we consider solving (a)-(c) sequentially.

$(a):\quad \mathbb{E}\left[(x^\top\beta)^2\right] = d.$

$(b):\quad \mathbb{E}\left[(\boldsymbol{x}^\top \boldsymbol{W}\boldsymbol{X}^\top \boldsymbol{X}\boldsymbol{\beta})^2\right]$

$$= \mathbb{E}\left[\left(\boldsymbol{x}^\top \boldsymbol{W}\sum_{i=1}^{n}(\alpha\boldsymbol{x} + \gamma\boldsymbol{g}_i)(\alpha\boldsymbol{x} + \gamma\boldsymbol{g}_i)^\top \boldsymbol{\beta}\right)^2\right]$$

$$= \mathbb{E}\left[\left(\sum_{i=1}^{n}\boldsymbol{x}^\top \boldsymbol{W}(\alpha^2\boldsymbol{x}\boldsymbol{x}^\top + \gamma^2\boldsymbol{g}_i\boldsymbol{g}_i^\top + \alpha\gamma\boldsymbol{x}\boldsymbol{g}_i^\top + \alpha\gamma\boldsymbol{g}_i\boldsymbol{x}^\top)\boldsymbol{\beta}\right)^2\right]$$

$$= \alpha^4 n^2 \mathbb{E}\left[(\boldsymbol{x}^\top \boldsymbol{W}\boldsymbol{x}\boldsymbol{x}^\top \boldsymbol{\beta})^2\right] + \gamma^4 \mathbb{E}\left[\left(\sum_{i=1}^{n}\boldsymbol{x}^\top \boldsymbol{W}\boldsymbol{g}_i\boldsymbol{g}_i^\top \boldsymbol{\beta}\right)^2\right] + \alpha^2\gamma^2 \mathbb{E}\left[\left(\sum_{i=1}^{n}\boldsymbol{x}^\top \boldsymbol{W}\boldsymbol{x}\boldsymbol{g}_i^\top \boldsymbol{\beta}\right)^2\right] + \alpha^2\gamma^2 \mathbb{E}\left[\left(\sum_{i=1}^{n}\boldsymbol{x}^\top \boldsymbol{W}\boldsymbol{g}_i\boldsymbol{x}^\top \boldsymbol{\beta}\right)^2\right]$$

$$\quad + 2\alpha^2\gamma^2 n^2 \mathbb{E}\left[\boldsymbol{x}^\top \boldsymbol{W}\boldsymbol{x}\boldsymbol{x}^\top \boldsymbol{\beta}\boldsymbol{\beta}^\top \boldsymbol{g}\boldsymbol{g}^\top \boldsymbol{W}^\top \boldsymbol{x}\right] + 2\alpha^2\gamma^2 n \mathbb{E}\left[\boldsymbol{x}^\top \boldsymbol{W}\boldsymbol{x}\boldsymbol{g}^\top \boldsymbol{\beta}\boldsymbol{x}^\top \boldsymbol{W}\boldsymbol{g}\boldsymbol{x}^\top \boldsymbol{\beta}\right]$$

$$= \left(\alpha^4 n^2(d+4)N_1 + \gamma^4 n(d+n+1)N_2\right) + \left(\alpha^2\gamma^2 ndN_1 + \alpha^2\gamma^2 n(d+2)N_2\right) + \left(2\alpha^2\gamma^2 n^2 N_1 + 2\alpha^2\gamma^2 nN_1\right)$$

$$= \left(\alpha^4 n^2(d+4) + \alpha^2\gamma^2 n(2n+d+2)\right)N_1 + \left(\alpha^2\gamma^2 n(d+2) + \gamma^4 n(d+n+1)\right)N_2$$

$$= A_1 N_1 + A_2 N_2.$$

$(c):\quad \mathbb{E}\left[\boldsymbol{x}^\top \boldsymbol{\beta}\boldsymbol{x}^\top \boldsymbol{W}\boldsymbol{X}^\top \boldsymbol{X}\boldsymbol{\beta}\right] = \mathbb{E}\left[\sum_{i=1}^{n}\boldsymbol{x}^\top \boldsymbol{\beta}\boldsymbol{x}^\top \boldsymbol{W}(\alpha\boldsymbol{x} + \gamma\boldsymbol{g}_i)(\alpha\boldsymbol{x} + \gamma\boldsymbol{g}_i)^\top \boldsymbol{\beta}\right]$

$$= \mathbb{E}\left[\sum_{i=1}^{n}\boldsymbol{x}^\top \boldsymbol{\beta}\boldsymbol{x}^\top \boldsymbol{W}(\alpha^2\boldsymbol{x}\boldsymbol{x}^\top + \gamma^2\boldsymbol{g}_i\boldsymbol{g}_i^\top + \alpha\gamma\boldsymbol{x}\boldsymbol{g}_i^\top + \alpha\gamma\boldsymbol{g}_i\boldsymbol{x}^\top)\boldsymbol{\beta}\right]$$

$$= \alpha^2 n \mathbb{E}\left[\boldsymbol{x}^\top \boldsymbol{\beta}\boldsymbol{x}^\top \boldsymbol{W}\boldsymbol{x}\boldsymbol{x}^\top \boldsymbol{\beta}\right] + \gamma^2 n \mathbb{E}\left[\boldsymbol{x}^\top \boldsymbol{\beta}\boldsymbol{x}^\top \boldsymbol{W}\boldsymbol{g}\boldsymbol{g}^\top \boldsymbol{\beta}\right]$$

$$= \alpha^2 n(d+2)\mathrm{tr}\,(\boldsymbol{W}) + \gamma^2 n\,\mathrm{tr}\,(\boldsymbol{W})$$

$$= \left(\alpha^2 n(d+2) + \gamma^2 n\right)N_3$$

$$= A_3 N_3.$$

Here, $(b)$ utilizes the 4'th and 6'th moment results (27) and (30) and we define

$$A_1 = \alpha^4 n^2(d+4) + \alpha^2\gamma^2 n(2n+d+2)$$
$$A_2 = \alpha^2\gamma^2 n(d+2) + \gamma^4 n(d+n+1)$$
$$A_3 = \alpha^2 n(d+2) + \gamma^2 n.$$

Then combining all together results in

$$\mathcal{L}(\boldsymbol{W}) = A_1 N_1 + A_2 N_2 - 2A_3 N_3 + n\sigma^2(\alpha^2 N_1 + \gamma^2 N_2) + d + \sigma^2.$$

To find the optimal solution, set $\nabla\mathcal{L}(\boldsymbol{W}) = 0$ and we obtain

$$A_1 \nabla N_1 + A_2 \nabla N_2 - 2A_3 \nabla N_3 + n\sigma^2(\alpha^2 \nabla N_1 + \gamma^2 \nabla N_2) = 0. \tag{39}$$

Note that we have

$$\nabla N_1 = \nabla\left(\mathrm{tr}\,(\boldsymbol{W})^2 + \mathrm{tr}\left(\boldsymbol{W}\boldsymbol{W}^\top\right) + \mathrm{tr}\left(\boldsymbol{W}^2\right)\right) = 2\mathrm{tr}\,(\boldsymbol{W})\boldsymbol{I} + 2\boldsymbol{W} + 2\boldsymbol{W}^\top$$
$$\nabla N_2 = \nabla\mathrm{tr}\left(\boldsymbol{W}\boldsymbol{W}^\top\right) = 2\boldsymbol{W}$$
$$\nabla N_3 = \nabla\mathrm{tr}\,(\boldsymbol{W}) = \boldsymbol{I}.$$

Therefore, (39) returns

$$2A_1\left(\mathrm{tr}\,(\boldsymbol{W})\boldsymbol{I} + \boldsymbol{W} + \boldsymbol{W}^\top\right) + 2A_2\boldsymbol{W} - 2A_3 + 2n\sigma^2(\alpha^2(\mathrm{tr}\,(\boldsymbol{W})\boldsymbol{I} + \boldsymbol{W} + \boldsymbol{W}^\top) + \gamma^2\boldsymbol{W})\boldsymbol{I} = 0, \tag{40}$$

which implies that the optimal solution $\boldsymbol{W}_\star$ has the form of $c\boldsymbol{I}$ for some constant $c$. Then suppose $\boldsymbol{W}_\star = c\boldsymbol{I}$, we have $\mathrm{tr}\,(\boldsymbol{W}) = cd$ and (40) returns

$$2A_1(d+2)c\boldsymbol{I} + 2A_2 c\boldsymbol{I} - 2A_3\boldsymbol{I} + 2n\sigma^2(\alpha^2(d+2)c\boldsymbol{I} + \gamma^2 c\boldsymbol{I}) = 0$$

$$\Longrightarrow c = \frac{A_3}{A_1(d+2) + A_2 + n\sigma^2(\alpha^2(d+2) + \gamma^2)}$$

$$= \frac{\alpha^2(d+2) + \gamma^2}{\alpha^4 n(d+2)(d+4) + \alpha^2\gamma^2(d+2)(d+2n+3) + \gamma^4(d+n+1) + \sigma^2(\alpha^2(d+2) + \gamma^2)}.$$

Then the optimal loss is obtained by setting $W_\star = cI$ and

$$\mathcal{L}_\star = \mathcal{L}(W_\star) = A_1 c^2 d(d+2) + A_2 c^2 d - 2A_3 cd + n\sigma^2 c^2 d(\alpha^2(d+2) + \gamma^2) + d + \sigma^2$$

$$= c^2 d \left( A_1(d+2) + A_2 + n\sigma^2(\alpha^2(d+2) + \gamma^2) \right) - 2A_3 cd + d + \sigma^2$$

$$= d + \sigma^2 - A_3 cd.$$

It completes the proof of (36). Now if assuming $\alpha = O\left(1/\sqrt{d}\right)$, $d/n = O(1)$ and sufficiently large dimension $d$, we have the approximate

$$c \approx \frac{\alpha^2 d + 1}{\alpha^4 d^2 n + \alpha^2 d(d+2n) + (d+n) + \sigma^2(\alpha^2 d + 1)}$$

$$= \frac{\alpha^2 d + 1}{(\alpha^2 d + 1)^2 n + (\alpha^2 d + 1)d + \sigma^2(\alpha^2 d + 1)}$$

$$= \frac{1}{(\alpha^2 d + 1)n + d + \sigma^2}$$

and

$$\mathcal{L}_\star \approx d + \sigma^2 - \frac{(\alpha^2 d + 1)nd}{(\alpha^2 d + 1)n + d + \sigma^2}.$$

$\square$

### C.4. Task-feature Alignment with $\alpha$ Correlation

In this section, we consider the task-feature alignment data model similar to (11), where we first draw task vector $\beta$ via

$$\beta \sim \mathcal{N}(0, I).$$

Then we generate examples $(x_i, y_i)_{i=1}^{n+1}$ according to the rule corr_coef$(x_i, \beta) \geq \alpha \geq 0$ via

$$x_i \,|\, \beta \sim \mathcal{N}(\alpha\beta, I), \quad \xi_i \sim \mathcal{N}(0, \sigma^2) \quad \text{and} \quad y_i = \gamma \cdot x_i^\top \beta + \xi_i, \tag{41}$$

which results in (11) by setting $\gamma^2 = 1/(\alpha^2 d + 1)$.

**Theorem C.2** (Extended version of Theorem 3.4)**.** *Consider linear model as defined in* (41)*. Recap the objective from* (5a) *and let* $W_\star := \arg\min_W \mathcal{L}_{PGD}(W)$*, and* $\mathcal{L}_\star = \mathcal{L}_{PGD}(W_\star)$*. Then* $W_\star$ *and* $\mathcal{L}_\star$ *satisfy*

$$W_\star = cI \qquad and \qquad \mathcal{L}_\star = d\gamma^2(\Delta_0 \alpha^2 + 1) + \sigma^2 - cnd\gamma^2(\Delta_1 \alpha^4 + 2\Delta_0 \alpha^2 + 1) \tag{42}$$

*where*

$$c = \frac{\Delta_1 \alpha^4 + 2\Delta_0 \alpha^2 + 1}{\Delta_2 \alpha^6 + \Delta_3 \alpha^4 + \Delta_4 \alpha^2 + (d+n+1) + \sigma^2(\Delta_0 \alpha^4 + 2\alpha^2 + 1)/\gamma^2}$$

*and*

$$\begin{cases} \Delta_0 = d + 2 \\ \Delta_1 = (d+2)(d+4) \\ \Delta_2 = (d+2)(d+4)(d+6)n \\ \Delta_3 = (d+2)(d+4)(3n+4) \\ \Delta_4 = (d+2)(3n+d+3) + (d+8). \end{cases}$$

*Suppose* $\alpha = O\left(1/\sqrt{d}\right)$*,* $d/n = O(1)$ *and* $d$ *is sufficiently large. Let* $\kappa = \alpha^2 d + 1$ *and* $\gamma^2 = 1/\kappa$*. Then* $W_\star$ *and* $\mathcal{L}_\star$ *have approximate forms*

$$W_\star \approx \frac{1}{\kappa n + (d+\sigma^2)/\kappa} \qquad and \qquad \mathcal{L}_\star \approx d + \sigma^2 - \frac{\kappa nd}{\kappa n + (d+\sigma^2)/\kappa}. \tag{43}$$

*Proof.* Here, for clean notation and without loss of generality, we define and rewrite (41) via

$$\boldsymbol{g}_i \sim \mathcal{N}(0, \boldsymbol{I}), \quad \xi_i \sim \mathcal{N}(0, \sigma^2) \quad \text{and} \quad \boldsymbol{x}_i = \alpha\boldsymbol{\beta} + \boldsymbol{g}_i, \quad y_i = \gamma\boldsymbol{x}_i^\top\boldsymbol{\beta} + \xi_i = \gamma \cdot (\alpha\boldsymbol{\beta} + \boldsymbol{g}_i)^\top\boldsymbol{\beta} + \xi_i.$$

Recap the loss function from (5a), we obtain

$$
\begin{aligned}
\mathcal{L}(\boldsymbol{W}) &= \mathbb{E}\left[(y - g(\boldsymbol{Z}))^2\right] \\
&= \mathbb{E}\left[\left(\gamma\boldsymbol{x}^\top\boldsymbol{\beta} + \xi - \boldsymbol{x}^\top\boldsymbol{W}\boldsymbol{X}^\top(\gamma\boldsymbol{X}\boldsymbol{\beta} + \boldsymbol{\xi})\right)^2\right] \\
&= \mathbb{E}\left[\gamma^2(\boldsymbol{x}^\top\boldsymbol{\beta} - \boldsymbol{x}^\top\boldsymbol{W}\boldsymbol{X}^\top\boldsymbol{X}\boldsymbol{\beta})^2 + 2\gamma(\boldsymbol{x}^\top\boldsymbol{\beta} - \boldsymbol{x}^\top\boldsymbol{W}\boldsymbol{X}^\top\boldsymbol{X}\boldsymbol{\beta})(\xi - \boldsymbol{x}^\top\boldsymbol{W}\boldsymbol{X}^\top\boldsymbol{\xi}) + (\xi - \boldsymbol{x}^\top\boldsymbol{W}\boldsymbol{X}^\top\boldsymbol{\xi})^2\right] \\
&= \gamma^2 \mathbb{E}\left[(\boldsymbol{x}^\top\boldsymbol{\beta} - \boldsymbol{x}^\top\boldsymbol{W}\boldsymbol{X}^\top\boldsymbol{X}\boldsymbol{\beta})^2\right] + \mathbb{E}\left[(\boldsymbol{x}^\top\boldsymbol{W}\boldsymbol{X}^\top\boldsymbol{\xi})^2\right] + \sigma^2.
\end{aligned}
\tag{44}
$$

Similar to Appendix C.3, to begin with, let

$$N_1 = \operatorname{tr}(\boldsymbol{W})^2 + \operatorname{tr}(\boldsymbol{W}\boldsymbol{W}^\top) + \operatorname{tr}(\boldsymbol{W}^2), \quad N_2 = \operatorname{tr}(\boldsymbol{W}\boldsymbol{W}^\top), \quad \text{and} \quad N_3 = \operatorname{tr}(\boldsymbol{W}),$$

and additionally, given $\boldsymbol{\Lambda}_{\boldsymbol{W}} = \boldsymbol{W} \odot \boldsymbol{I}$, let

$$N_4 = 3\operatorname{tr}(\boldsymbol{\Lambda}_{\boldsymbol{W}}^2) + (d + 4)\operatorname{tr}(\boldsymbol{W}\boldsymbol{W}^\top) + \operatorname{tr}(\boldsymbol{W}^2).$$

We first focus on the second term in (44)

$$
\begin{aligned}
\mathbb{E}\left[(\boldsymbol{x}^\top\boldsymbol{W}\boldsymbol{X}^\top\boldsymbol{\xi})^2\right] &= \mathbb{E}\left[\left((\alpha\boldsymbol{\beta} + \boldsymbol{g})^\top\boldsymbol{W}\sum_{i=1}^n \xi_i(\alpha\boldsymbol{\beta} + \boldsymbol{g}_i)\right)^2\right] \\
&= n\sigma^2 \mathbb{E}\left[\left((\alpha\boldsymbol{\beta} + \boldsymbol{g})^\top\boldsymbol{W}(\alpha\boldsymbol{\beta} + \boldsymbol{g}')\right)^2\right] \\
&= n\sigma^2 \left(\alpha^4 \mathbb{E}\left[(\boldsymbol{\beta}^\top\boldsymbol{W}\boldsymbol{\beta})^2\right] + 2\alpha^2 \mathbb{E}\left[(\boldsymbol{\beta}^\top\boldsymbol{W}\boldsymbol{g}')^2\right] + \mathbb{E}\left[(\boldsymbol{g}^\top\boldsymbol{W}\boldsymbol{g}')^2\right]\right) \\
&= n\sigma^2 \left(\alpha^4 \left(\operatorname{tr}(\boldsymbol{W})^2 + \operatorname{tr}(\boldsymbol{W}^2) + \operatorname{tr}(\boldsymbol{W}\boldsymbol{W}^\top)\right) + (2\alpha^2 + 1)\operatorname{tr}(\boldsymbol{W}\boldsymbol{W}^\top)\right) \\
&= n\sigma^2 \left(\alpha^4 N_1 + (2\alpha^2 + 1)N_2\right). \qquad \text{(It follows (27) and independence of } \boldsymbol{\beta}, \boldsymbol{g}, \boldsymbol{g}'.)
\end{aligned}
$$

Next, the first term of (44) (omitting $\gamma^2$) returns the following decomposition:

$$
\begin{aligned}
\mathbb{E}\left[(\boldsymbol{x}^\top\boldsymbol{\beta} - \boldsymbol{x}^\top\boldsymbol{W}\boldsymbol{X}^\top\boldsymbol{X}\boldsymbol{\beta})^2\right] &= \mathbb{E}\left[((\alpha\boldsymbol{\beta} + \boldsymbol{g})^\top(\boldsymbol{\beta} - \boldsymbol{W}\boldsymbol{X}^\top\boldsymbol{X}\boldsymbol{\beta}))^2\right] \\
&= \mathbb{E}\left[\left(\alpha\boldsymbol{\beta}^\top\boldsymbol{\beta} - \alpha\boldsymbol{\beta}^\top\boldsymbol{W}\boldsymbol{X}^\top\boldsymbol{X}\boldsymbol{\beta} + \boldsymbol{g}^\top\boldsymbol{\beta} - \boldsymbol{g}^\top\boldsymbol{W}\boldsymbol{X}^\top\boldsymbol{X}\boldsymbol{\beta}\right)^2\right] \\
&= \alpha^2 \mathbb{E}[(\boldsymbol{\beta}^\top\boldsymbol{\beta})^2] + \alpha^2 \mathbb{E}[(\boldsymbol{\beta}^\top\boldsymbol{W}\boldsymbol{X}^\top\boldsymbol{X}\boldsymbol{\beta})^2] + \mathbb{E}[(\boldsymbol{g}^\top\boldsymbol{\beta})^2] + \mathbb{E}[(\boldsymbol{g}^\top\boldsymbol{W}\boldsymbol{X}^\top\boldsymbol{X}\boldsymbol{\beta})^2] \\
&\quad - 2\alpha^2 \mathbb{E}[\boldsymbol{\beta}^\top\boldsymbol{\beta}\boldsymbol{\beta}^\top\boldsymbol{W}\boldsymbol{X}^\top\boldsymbol{X}\boldsymbol{\beta}] - 2\mathbb{E}[\boldsymbol{\beta}^\top\boldsymbol{g}\boldsymbol{g}^\top\boldsymbol{W}\boldsymbol{X}^\top\boldsymbol{X}\boldsymbol{\beta}] \\
&= \alpha^2 d(d + 2) + \alpha^2 \underbrace{\mathbb{E}[(\boldsymbol{\beta}^\top\boldsymbol{W}\boldsymbol{X}^\top\boldsymbol{X}\boldsymbol{\beta})^2]}_{(a)} + d + \underbrace{\mathbb{E}[(\boldsymbol{g}^\top\boldsymbol{W}\boldsymbol{X}^\top\boldsymbol{X}\boldsymbol{\beta})^2]}_{(b)} \\
&\quad - 2\alpha^2 \underbrace{\mathbb{E}[\boldsymbol{\beta}^\top\boldsymbol{\beta}\boldsymbol{\beta}^\top\boldsymbol{W}\boldsymbol{X}^\top\boldsymbol{X}\boldsymbol{\beta}]}_{(c)} - 2\underbrace{\mathbb{E}[\boldsymbol{\beta}^\top\boldsymbol{g}\boldsymbol{g}^\top\boldsymbol{W}\boldsymbol{X}^\top\boldsymbol{X}\boldsymbol{\beta}]}_{(d)}.
\end{aligned}
$$

Consider solving (*a*)-(*d*) sequentially as follows:

To begin with, we use the following decomposition for all (*a*)-(*d*):

$$
\begin{aligned}
\boldsymbol{X}^\top\boldsymbol{X}\boldsymbol{\beta} &= \sum_{i=1}^n \boldsymbol{x}_i\boldsymbol{x}_i^\top\boldsymbol{\beta} \\
&= \sum_{i=1}^n (\alpha\boldsymbol{\beta} + \boldsymbol{g}_i)(\alpha\boldsymbol{\beta} + \boldsymbol{g}_i)^\top\boldsymbol{\beta} \\
&= \sum_{i=1}^n \alpha^2\boldsymbol{\beta}\boldsymbol{\beta}^\top\boldsymbol{\beta} + \alpha\boldsymbol{\beta}\boldsymbol{g}_i^\top\boldsymbol{\beta} + \alpha\boldsymbol{g}_i\boldsymbol{\beta}^\top\boldsymbol{\beta} + \boldsymbol{g}_i\boldsymbol{g}_i^\top\boldsymbol{\beta}.
\end{aligned}
$$

Then, we have

$$(a): \quad \mathbb{E}[(\boldsymbol{\beta}^\top \boldsymbol{W} \boldsymbol{X}^\top \boldsymbol{X} \boldsymbol{\beta})^2]$$

$$= \mathbb{E}\left[\left(\sum_{i=1}^n \alpha^2 \boldsymbol{\beta}^\top \boldsymbol{W} \boldsymbol{\beta} \boldsymbol{\beta}^\top \boldsymbol{\beta} + \alpha \boldsymbol{\beta}^\top \boldsymbol{W} \boldsymbol{\beta} \boldsymbol{g}_i^\top \boldsymbol{\beta} + \alpha \boldsymbol{\beta}^\top \boldsymbol{W} \boldsymbol{g}_i \boldsymbol{\beta}^\top \boldsymbol{\beta} + \boldsymbol{\beta}^\top \boldsymbol{W} \boldsymbol{g}_i \boldsymbol{g}_i^\top \boldsymbol{\beta}\right)^2\right]$$

$$= \alpha^4 n^2 \mathbb{E}\left[\left(\boldsymbol{\beta}^\top \boldsymbol{W} \boldsymbol{\beta} \boldsymbol{\beta}^\top \boldsymbol{\beta}\right)^2\right] + \alpha^2 \mathbb{E}\left[\left(\sum_{i=1}^n \boldsymbol{\beta}^\top \boldsymbol{W} \boldsymbol{\beta} \boldsymbol{g}_i^\top \boldsymbol{\beta}\right)^2\right] + \alpha^2 \mathbb{E}\left[\left(\sum_{i=1}^n \boldsymbol{\beta}^\top \boldsymbol{W} \boldsymbol{g}_i \boldsymbol{\beta}^\top \boldsymbol{\beta}\right)^2\right] + \mathbb{E}\left[\left(\sum_{i=1}^n \boldsymbol{\beta}^\top \boldsymbol{W} \boldsymbol{g}_i \boldsymbol{g}_i^\top \boldsymbol{\beta}\right)^2\right]$$

$$+ 2\alpha^2 n \mathbb{E}\left[\sum_{i=1}^n \boldsymbol{\beta}^\top \boldsymbol{W} \boldsymbol{\beta} \boldsymbol{\beta}^\top \boldsymbol{\beta} \boldsymbol{\beta}^\top \boldsymbol{W} \boldsymbol{g}_i \boldsymbol{g}_i^\top \boldsymbol{\beta}\right] + 2\alpha^2 \mathbb{E}\left[\sum_{i=1}^n \boldsymbol{\beta}^\top \boldsymbol{W} \boldsymbol{\beta} \boldsymbol{g}_i^\top \boldsymbol{\beta} \boldsymbol{\beta}^\top \boldsymbol{W} \boldsymbol{g}_i \boldsymbol{\beta}^\top \boldsymbol{\beta}\right]$$

$$= \alpha^4 n^2 \mathbb{E}\left[\left(\boldsymbol{\beta}^\top \boldsymbol{W} \boldsymbol{\beta} \boldsymbol{\beta}^\top \boldsymbol{\beta}\right)^2\right] + \alpha^2 n \mathbb{E}\left[\left(\boldsymbol{\beta}^\top \boldsymbol{W} \boldsymbol{\beta} \boldsymbol{g}'^\top \boldsymbol{\beta}\right)^2\right] + \alpha^2 n \mathbb{E}\left[\left(\boldsymbol{\beta}^\top \boldsymbol{W} \boldsymbol{g}' \boldsymbol{\beta}^\top \boldsymbol{\beta}\right)^2\right] + \mathbb{E}\left[\left(\sum_{i=1}^n \boldsymbol{\beta}^\top \boldsymbol{W} \boldsymbol{g}_i \boldsymbol{g}_i^\top \boldsymbol{\beta}\right)^2\right]$$

$$+ 2\alpha^2 n^2 \mathbb{E}\left[\boldsymbol{\beta}^\top \boldsymbol{W} \boldsymbol{\beta} \boldsymbol{\beta}^\top \boldsymbol{\beta} \boldsymbol{\beta}^\top \boldsymbol{W} \boldsymbol{g}' \boldsymbol{g}'^\top \boldsymbol{\beta}\right] + 2\alpha^2 n \mathbb{E}\left[\boldsymbol{\beta}^\top \boldsymbol{W} \boldsymbol{\beta} \boldsymbol{g}_i^\top \boldsymbol{\beta} \boldsymbol{\beta}^\top \boldsymbol{W} \boldsymbol{g}_i \boldsymbol{\beta}^\top \boldsymbol{\beta}\right]$$

$$= \alpha^4 n^2 (d+4)(d+6)N_1 + \alpha^2 n(d+4)N_1 + \alpha^2 n(d+2)(d+4)N_2 \tag{45}$$

$$+ n(n-1)N_1 + nN_4 \tag{46}$$

$$+ 2\alpha^2 n^2 (d+4)N_1 + 2\alpha^2 n(d+4)N_1 \tag{47}$$

$$= \left(\alpha^2 n(d+4)(\alpha^2 n(d+6) + 2n + 3) + n(n-1)\right)N_1 + \alpha^2 n(d+2)(d+4)N_2 + nN_4 \tag{48}$$

$$= B_1 N_1 + B_2 N_2 + nN_4,$$

where (45) and (47) utilize (30) and (32), and (46) is obtained via

$$\mathbb{E}\left[\left(\sum_{i=1}^n \boldsymbol{\beta}^\top \boldsymbol{W} \boldsymbol{g}_i \boldsymbol{g}_i^\top \boldsymbol{\beta}\right)^2\right] = n \mathbb{E}\left[\left(\boldsymbol{\beta}^\top \boldsymbol{W} \boldsymbol{g}' \boldsymbol{g}'^\top \boldsymbol{\beta}\right)^2\right] + n(n-1) \mathbb{E}\left[\boldsymbol{\beta}^\top \boldsymbol{W} \boldsymbol{g}' \boldsymbol{g}'^\top \boldsymbol{\beta} \boldsymbol{\beta}^\top \boldsymbol{W} \boldsymbol{g}'' \boldsymbol{g}''^\top \boldsymbol{\beta}\right]$$

$$= nN_4 + n(n-1)N_1,$$

which follows (27) and (28).

$$(b): \quad \mathbb{E}\left[(\boldsymbol{g}^\top \boldsymbol{W} \boldsymbol{X}^\top \boldsymbol{X} \boldsymbol{\beta})^2\right]$$

$$= \mathbb{E}\left[\left(\sum_{i=1}^n \alpha^2 \boldsymbol{g}^\top \boldsymbol{W} \boldsymbol{\beta} \boldsymbol{\beta}^\top \boldsymbol{\beta} + \alpha \boldsymbol{g}^\top \boldsymbol{W} \boldsymbol{\beta} \boldsymbol{g}_i^\top \boldsymbol{\beta} + \alpha \boldsymbol{g}^\top \boldsymbol{W} \boldsymbol{g}_i \boldsymbol{\beta}^\top \boldsymbol{\beta} + \boldsymbol{g}^\top \boldsymbol{W} \boldsymbol{g}_i \boldsymbol{g}_i^\top \boldsymbol{\beta}\right)^2\right]$$

$$= \alpha^4 n^2 \mathbb{E}\left[\left(\boldsymbol{g}^\top \boldsymbol{W} \boldsymbol{\beta} \boldsymbol{\beta}^\top \boldsymbol{\beta}\right)^2\right] + \alpha^2 \mathbb{E}\left[\left(\sum_{i=1}^n \boldsymbol{g}^\top \boldsymbol{W} \boldsymbol{\beta} \boldsymbol{g}_i^\top \boldsymbol{\beta}\right)^2\right] + \alpha^2 \mathbb{E}\left[\left(\sum_{i=1}^n \boldsymbol{g}^\top \boldsymbol{W} \boldsymbol{g}_i \boldsymbol{\beta}^\top \boldsymbol{\beta}\right)^2\right] + \mathbb{E}\left[\left(\sum_{i=1}^n \boldsymbol{g}^\top \boldsymbol{W} \boldsymbol{g}_i \boldsymbol{g}_i^\top \boldsymbol{\beta}\right)^2\right]$$

$$+ 2\alpha^2 n \mathbb{E}\left[\sum_{i=1}^n \boldsymbol{g}^\top \boldsymbol{W} \boldsymbol{\beta} \boldsymbol{\beta}^\top \boldsymbol{\beta} \boldsymbol{g}^\top \boldsymbol{W} \boldsymbol{g}_i \boldsymbol{g}_i^\top \boldsymbol{\beta}\right] + 2\alpha^2 \mathbb{E}\left[\sum_{i=1}^n \boldsymbol{g}^\top \boldsymbol{W} \boldsymbol{\beta} \boldsymbol{g}_i^\top \boldsymbol{\beta} \boldsymbol{g}^\top \boldsymbol{W} \boldsymbol{g}_i \boldsymbol{\beta}^\top \boldsymbol{\beta}\right]$$

$$= \alpha^4 n^2 \mathbb{E}\left[\left(\boldsymbol{g}^\top \boldsymbol{W} \boldsymbol{\beta} \boldsymbol{\beta}^\top \boldsymbol{\beta}\right)^2\right] + \alpha^2 n \mathbb{E}\left[\left(\boldsymbol{g}^\top \boldsymbol{W} \boldsymbol{\beta} \boldsymbol{g}'^\top \boldsymbol{\beta}\right)^2\right] + \alpha^2 n \mathbb{E}\left[\left(\boldsymbol{g}^\top \boldsymbol{W} \boldsymbol{g}' \boldsymbol{\beta}^\top \boldsymbol{\beta}\right)^2\right] + \mathbb{E}\left[\left(\sum_{i=1}^n \boldsymbol{g}^\top \boldsymbol{W} \boldsymbol{g}_i \boldsymbol{g}_i^\top \boldsymbol{\beta}\right)^2\right]$$

$$+ 2\alpha^2 n^2 \mathbb{E}\left[\boldsymbol{g}^\top \boldsymbol{W} \boldsymbol{\beta} \boldsymbol{\beta}^\top \boldsymbol{\beta} \boldsymbol{g}^\top \boldsymbol{W} \boldsymbol{g}' \boldsymbol{g}'^\top \boldsymbol{\beta}\right] + 2\alpha^2 n \mathbb{E}\left[\boldsymbol{g}^\top \boldsymbol{W} \boldsymbol{\beta} \boldsymbol{g}_i^\top \boldsymbol{\beta} \boldsymbol{g}^\top \boldsymbol{W} \boldsymbol{g}_i \boldsymbol{\beta}^\top \boldsymbol{\beta}\right]$$

$$= \alpha^4 n^2 (d+2)(d+4)N_2 + \alpha^2 n(d+2)N_2 + \alpha^2 nd(d+2)N_2 + n(d+n+1)N_2 \tag{49}$$

$$+ 2\alpha^2 n^2 (d+2)N_2 + 2\alpha^2 n(d+2)N_2 \tag{50}$$

$$= \left(\alpha^2 n(d+2)(\alpha^2 n(d+4) + 2n + d + 3) + n(d+n-1)\right)N_2$$

$$= B_3 N_2,$$

where (49) and (50) are obtained using (27), (30) and

$$\mathbb{E}\left[\left(\sum_{i=1}^n \boldsymbol{g}^\top \boldsymbol{W} \boldsymbol{g}_i \boldsymbol{g}_i^\top \boldsymbol{\beta}\right)^2\right] = n \mathbb{E}\left[\left(\boldsymbol{g}^\top \boldsymbol{W} \boldsymbol{g}' \boldsymbol{g}'^\top \boldsymbol{\beta}\right)^2\right] + n(n-1) \mathbb{E}\left[\boldsymbol{g}^\top \boldsymbol{W} \boldsymbol{g}' \boldsymbol{g}'^\top \boldsymbol{\beta} \boldsymbol{g}^\top \boldsymbol{W} \boldsymbol{g}'' \boldsymbol{g}''^\top \boldsymbol{\beta}\right]$$

$$= n(d+2)N_2 + n(n-1)N_2 = n(n+d+1)N_2.$$

$(c):\quad \mathbb{E}\left[\boldsymbol{\beta}^\top \boldsymbol{\beta}\boldsymbol{\beta}^\top \boldsymbol{W}\boldsymbol{X}^\top \boldsymbol{X}\boldsymbol{\beta}\right]$

$\qquad = n\,\mathbb{E}\left[\boldsymbol{\beta}^\top \boldsymbol{\beta}\boldsymbol{\beta}^\top \boldsymbol{W}(\alpha\boldsymbol{\beta} + \boldsymbol{g}')(\alpha\boldsymbol{\beta} + \boldsymbol{g}')^\top \boldsymbol{\beta}\right]$

$\qquad = \alpha^2 n\,\mathbb{E}\left[\boldsymbol{\beta}^\top \boldsymbol{\beta}\boldsymbol{\beta}^\top \boldsymbol{W}\boldsymbol{\beta}\boldsymbol{\beta}^\top \boldsymbol{\beta}\right] + n\,\mathbb{E}\left[\boldsymbol{\beta}^\top \boldsymbol{\beta}\boldsymbol{\beta}^\top \boldsymbol{W}\boldsymbol{g}'\boldsymbol{g}'^\top \boldsymbol{\beta}\right]$

$\qquad = \alpha^2 n(d+2)(d+4)\mathrm{tr}\,(\boldsymbol{W}) + n(d+2)\mathrm{tr}\,(\boldsymbol{W})$

$\qquad = \left(\alpha^2 n(d+2)(d+4) + n(d+2)\right)N_3$

$\qquad = B_4 N_3.$

$(d):\quad \mathbb{E}\left[\boldsymbol{\beta}^\top \boldsymbol{g}\boldsymbol{g}^\top \boldsymbol{W}\boldsymbol{X}^\top \boldsymbol{X}\boldsymbol{\beta}\right]$

$\qquad = n\,\mathbb{E}\left[\boldsymbol{\beta}^\top \boldsymbol{g}\boldsymbol{g}^\top \boldsymbol{W}(\alpha\boldsymbol{\beta} + \boldsymbol{g}')(\alpha\boldsymbol{\beta} + \boldsymbol{g}')^\top \boldsymbol{\beta}\right]$

$\qquad = \alpha^2 n\,\mathbb{E}\left[\boldsymbol{\beta}^\top \boldsymbol{g}\boldsymbol{g}^\top \boldsymbol{W}\boldsymbol{\beta}\boldsymbol{\beta}^\top \boldsymbol{\beta}\right] + n\,\mathbb{E}\left[\boldsymbol{\beta}^\top \boldsymbol{g}\boldsymbol{g}^\top \boldsymbol{W}\boldsymbol{g}'\boldsymbol{g}'^\top \boldsymbol{\beta}\right]$

$\qquad = \alpha^2 n(d+2)\mathrm{tr}\,(\boldsymbol{W}) + n\,\mathrm{tr}\,(\boldsymbol{W})$

$\qquad = \left(\alpha^2 n(d+2) + n\right)N_3$

$\qquad = B_5 N_3.$

Here we define

$$B_1 = \alpha^2 n(d+4)(\alpha^2 n(d+6) + 2n + 3) + n(n-1)$$

$$B_2 = \alpha^2 n(d+2)(d+4)$$

$$B_3 = \alpha^2 n(d+2)(\alpha^2 n(d+4) + 2n + d + 3) + n(d+n-1)$$

$$B_4 = \alpha^2 n(d+2)(d+4) + n(d+2)$$

$$B_5 = \alpha^2 n(d+2) + n.$$

Then combining all together results in

$$\mathcal{L}(\boldsymbol{W}) = \gamma^2\left(\alpha^2 d(d+2) + d + \alpha^2(B_1 N_1 + B_2 N_2 + n N_4) + B_3 N_2 - 2\alpha^2 B_4 N_3 - 2 B_5 N_3\right) + n\sigma^2(\alpha^4 N_1 + (2\alpha^2 + 1)N_2) + \sigma^2$$

$$= \gamma^2\left(\alpha^2 B_1 N_1 + (\alpha^2 B_2 + B_3)N_2 - 2(\alpha^2 B_4 + B_5)N_3 + \alpha^2 n N_4\right) + n\sigma^2(\alpha^4 N_1 + (2\alpha^2 + 1)N_2) + \gamma^2 d\left(\alpha^2(d+2) + 1\right) + \sigma^2$$

and differentiating it results in

$$\nabla\mathcal{L}(\boldsymbol{W}) = \gamma^2\left(\alpha^2 B_1 \nabla N_1 + (\alpha^2 B_2 + B_3)\nabla N_2 - 2(\alpha^2 B_4 + B_5)\nabla N_3 + \alpha^2 n \nabla N_4\right) + n\sigma^2(\alpha^4 \nabla N_1 + (2\alpha^2 + 1)\nabla N_2).$$

Similar to the proof in Appendix C.3, $\boldsymbol{W}_\star$ has the form of $\boldsymbol{W}_\star = c\boldsymbol{I}$ and we have

$$\nabla N_1 = \nabla\left(\mathrm{tr}\,(\boldsymbol{W})^2 + \mathrm{tr}\left(\boldsymbol{W}\boldsymbol{W}^\top\right) + \mathrm{tr}\left(\boldsymbol{W}^2\right)\right) = 2\mathrm{tr}\,(\boldsymbol{W})\,\boldsymbol{I} + 2\boldsymbol{W} + 2\boldsymbol{W}^\top = 2c(d+2)\boldsymbol{I}$$

$$\nabla N_2 = \nabla\mathrm{tr}\left(\boldsymbol{W}\boldsymbol{W}^\top\right) = 2\boldsymbol{W} = 2c\boldsymbol{I}$$

$$\nabla N_3 = \nabla\mathrm{tr}\,(\boldsymbol{W}) = \boldsymbol{I}$$

$$\nabla N_4 = \nabla\left(3\mathrm{tr}\left(\boldsymbol{\Lambda}_{\boldsymbol{W}}^2\right) + (d+4)\mathrm{tr}\left(\boldsymbol{W}\boldsymbol{W}^\top\right) + \mathrm{tr}\left(\boldsymbol{W}^2\right)\right)$$

$$\qquad = 6\cdot\mathrm{diag}\,(\boldsymbol{\Lambda}_{\boldsymbol{W}}) + 2(d+4)\boldsymbol{W} + 2\boldsymbol{W}^\top$$

$$\qquad = 2c(d+8)\boldsymbol{I}.$$

Therefore, setting $\nabla\mathcal{L}(\boldsymbol{W}) = 0$ returns

$$\gamma^2\left(2c(d+2)\alpha^2 B_1 + 2c(\alpha^2 B_2 + B_3) - 2(\alpha^2 B_4 + B_5) + 2c(d+8)\alpha^2 n\right) + 2cn\sigma^2(\alpha^4(d+2) + 2\alpha^2 + 1) = 0$$

$$\implies c = \frac{\alpha^2 B_4 + B_5}{(d+2)\alpha^2 B_1 + (\alpha^2 B_2 + B_3) + (d+8)\alpha^2 n + n\sigma^2(\alpha^4(d+2) + 2\alpha^2 + 1)/\gamma^2}$$

$$= \frac{\alpha^4 n(d+2)(d+4) + 2\alpha^2 n(d+2) + n}{\alpha^6 n^2(d+2)(d+4)(d+6) + \alpha^4 n(d+2)(d+4)(3n+4) + \alpha^2 n((d+2)(3n+d+3) + (d+8)) + n(d+n+1) + n\sigma^2(\alpha^4(d+2) + 2\alpha^2 + 1)/\gamma^2}$$

$$= \frac{\alpha^4(d+2)(d+4) + 2\alpha^2(d+2) + 1}{\alpha^6 n(d+2)(d+4)(d+6) + \alpha^4(d+2)(d+4)(3n+4) + \alpha^2((d+2)(3n+d+3) + (d+8)) + (d+n+1) + \sigma^2(\alpha^4(d+2) + 2\alpha^2 + 1)/\gamma^2}.$$

Then the optimal loss is obtained by setting $W_\star = cI$ and

$$\mathcal{L}_\star = \mathcal{L}(W_\star) = \gamma^2 d(\alpha^2(d+2)+1) + \sigma^2 - \gamma^2(\alpha^2 B_4 + B_5)cd.$$

It completes the proof of (42). Now if assuming $\alpha = O\left(1/\sqrt{d}\right), d/n = O(1), \gamma^2 = 1/(\alpha^2 d + 1)$ and sufficiently large dimension $d$, we have the approximate

$$c \approx \frac{\alpha^4 d^2 + 2\alpha^2 d + 1}{n\alpha^6 d^3 + 3n\alpha^4 d^2 + (3n+d)\alpha^2 d + d + n + \sigma^2(\alpha^4 d + 2\alpha^2 + 1)/\gamma^2}$$

$$\approx \frac{(\alpha^2 d + 1)^2}{n(\alpha^2 d + 1)^3 + d(\alpha^2 d + 1) + \sigma^2(\alpha^2 d + 1)}$$

$$\approx \frac{1}{(\alpha^2 d + 1)n + (d + \sigma^2)/(\alpha^2 d + 1)}$$

and

$$\mathcal{L}_\star \approx \gamma^2 d(\alpha^2 d + 1) + \sigma^2 - \frac{\gamma^2(\alpha^2 d + 1)^2 nd}{(\alpha^2 d + 1)n + (d + \sigma^2)/(\alpha^2 d + 1)}$$

$$= d + \sigma^2 - \frac{(\alpha^2 d + 1)nd}{(\alpha^2 d + 1)n + (d + \sigma^2)/(\alpha^2 d + 1)}.$$

$\square$

# D. Analysis of Low-Rank Parameterization

## D.1. Proof of Lemma 3.5

*Proof.* Recall the loss function from (34)

$$\mathcal{L}(W) = M - 2n\mathtt{tr}\left(\Sigma\bar{W}\right) + n(n+1)\mathtt{tr}\left(\Sigma\bar{W}^\top\bar{W}\right) + nM\mathtt{tr}\left(\bar{W}\bar{W}^\top\right)$$

where $\bar{W} = \Sigma_x^{1/2} W \Sigma_x^{1/2}, \Sigma = \Sigma_x^{1/2}\Sigma_\beta\Sigma_x^{1/2}$ and $M = \mathtt{tr}(\Sigma) + \sigma^2$. For any $\bar{W}$, let us parameterize $\bar{W} = UEU^\top$ where $U \in \mathbb{R}^{d\times r}$ denotes the eigenvectors of $\bar{W}$ and $E \in \mathbb{R}^{r\times r}$ is a symmetric square matrix. We will first treat $U$ as fixed and optimize $E$. We will then optimize $U$. Fixing $U$, setting $\bar{\Sigma} = U^\top\Sigma U$, we obtain

$$\mathcal{L}(E) = M - 2n\mathtt{tr}\left(\bar{\Sigma}E\right) + n(n+1)\mathtt{tr}\left(\bar{\Sigma}E^2\right) + nM\mathtt{tr}\left(E^2\right).$$

Differentiating, we obtain

$$0.5n^{-1}\nabla\mathcal{L}(E) = -\bar{\Sigma} + (n+1)\bar{\Sigma}E + ME.$$

Setting $\nabla\mathcal{L}(E) = 0$ returns

$$E_\star = (MI + (n+1)\bar{\Sigma})^{-1}\bar{\Sigma}. \tag{51}$$

Let $\bar{\lambda}_i$ denote the $i$'th largest eigenvalue of $\bar{\Sigma}$. Plugging in this value, we obtain the optimal risk as a function of $U$ is given by

$$\mathcal{L}_\star(U) = M - n\cdot\mathtt{tr}\left(\bar{\Sigma}E_\star\right) = M - n\cdot\mathtt{tr}\left((MI + (n+1)\bar{\Sigma})^{-1}\bar{\Sigma}^2\right) \tag{52}$$

$$= M - n\sum_{i=1}^r \frac{\bar{\lambda}_i^2}{(n+1)\bar{\lambda}_i + M} = M - n\sum_{i=1}^r \frac{\bar{\lambda}_i}{n+1+M\bar{\lambda}_i^{-1}}. \tag{53}$$

Now observe that, the right hand side is strictly decreasing function of the eigenvalues $\bar{\lambda}_i$ of $\bar{\Sigma} = U^\top\Sigma U$. Thus, to minimize $\mathcal{L}_\star(U)$, we need to maximize $\sum_{i=1}^r \frac{\bar{\lambda}_i}{n+1+M\bar{\lambda}_i^{-1}}$. It follows from Cauchy interlacing theorem that $\bar{\lambda}_j \leq \lambda_i$ where $\lambda_i$ is the $i$'th largest eigenvalue of $\Sigma$ since $\bar{\Sigma}$ is an orthogonal projection of $\Sigma$ on $U$. Consequently, we find the desired bound where

$$\mathcal{L}_\star = M - n\sum_{i=1}^r \frac{\lambda_i}{n+1+M\lambda_i^{-1}}.$$

The equality holds by setting $U$ to be the top-$r$ eigenvectors of $\Sigma$ and $E = E_\star(U)$ to be the diagonal matrix according to (51). $\square$

