# OpenReview forum: "Fine-grained Analysis of In-context Linear Estimation: Data, Architecture, and Beyond"
_ICML.cc/2024/Workshop/ICL — ICML 2024 Workshop ICL Poster_

### Official Review · Reviewer_EZys · 2024-06-07

**Rating:** 3
**Fit:** 3
**Confidence:** 2

**Workshop Review:**

I think the paper is solid, with many theoretical contributions. It connects ICL to PGD for linear attention and H3 models, and it also provides insights into RAG and LoRA. Some high-level experiments are conducted to validate the claims.

One simple suggestion is that when describing the results in Section 5, it is recommended not to just refer to the hyperlink pointing to the previous theorem. Instead, remind readers what the theorem suggests. This will make the paper easier to read.

**Reason For Not Giving Higher Score:**

NA

**Reason For Not Giving Lower Score:**

Solid paper with a lot of theoretical contribution.

---

### Official Review · Reviewer_qf2N · 2024-06-15
**Good paper.**

**Rating:** 2
**Fit:** 3
**Confidence:** 2

**Workshop Review:**

This paper studies the fine-grained properties of in-context learning of linear regression. In general, I think this paper is qualified for this workshop.

In this paper, the authors first studied the linear self-attention layer and the H3 layer and showed that they can both implement one step of pre-conditioned gradient descent (PGD). Furthermore, the H3 layer can implement Weighted PGD and is more powerful than LSA. They showed that the optimal ICL loss for PGD, LSA, and H3 layers are the same, showing that the optimal LSA or optimal H3 actually implements the PGD. They then analyze the case where x_i are correlated with each other by RAG and show that the positive correlation among covariates can help reduce the optimal ICL loss for linear regression. They show the optimal ICL loss under the restriction that W is of low rank. They finally did experiments to verify the effectiveness of their theory.

In general, this paper provides some new intuition for the ICL of linear regression, including the positive correlation among covariates and the case when facing the low-rank approximation. I think this paper is good and matches the topic of this workshop.

**Reason For Not Giving Higher Score:**

/

**Reason For Not Giving Lower Score:**

/

---

### Meta-Review · Area_Chair_JB6F · 2024-06-14

**Recommendation:** 2

**Metareview:**

This paper studies the landscape of one-layer linear attention and H3 models and prove that both models implement 1-step preconditioned gradient descent (PGD) and that the gating mechanism in H3 imitates attention. Authors further provide insights into Retrieval Augmented Generation (RAG) and show that under low-rank parametrication, optimal attention-weights still implement PGD.

The reviewer consider this paper interesting and with many theoretical contributions.

---

### Decision · Program_Chairs · 2024-06-17

Accept (Poster)